# Efa6 protects axons and regulates their growth and branching by inhibiting microtubule polymerisation at the cortex

Yue Qu[1][†], Ines Hahn[1][†]*, Meredith Lees[1], Jill Parkin[1], André Voelzmann[1], Karel Dorey[2], Alex Rathbone[3], Claire T Friel[3], Victoria J Allan[1], Pilar Okenve-Ramos[4], Natalia Sanchez-Soriano[4], Andreas Prokop[1]

[1]Manchester Academic Health Science Centre, Faculty of Biology, Medicine and Health, School of Biological Sciences, The University of Manchester, Manchester, United Kingdom; [2]Faculty of Biology, Medicine and Health, School of Medical Sciences, The University of Manchester, Manchester, United Kingdom; [3]School of Life Sciences, Faculty of Medicine and Health Sciences, The University of Nottingham, Nottingham, United Kingdom; [4]Department of Cellular and Molecular Physiology,Institute of Translational Medicine, University of Liverpool, Liverpool, United Kingdom

*For correspondence:
ines.hahn@manchester.ac.uk

[†]These authors contributed equally to this work

**Abstract** Cortical collapse factors affect microtubule (MT) dynamics at the plasma membrane. They play important roles in neurons, as suggested by inhibition of axon growth and regeneration through the ARF activator Efa6 in *C. elegans*, and by neurodevelopmental disorders linked to the mammalian kinesin Kif21A. How cortical collapse factors influence axon growth is little understood. Here we studied them, focussing on the function of *Drosophila* Efa6 in experimentally and genetically amenable fly neurons. First, we show that *Drosophila* Efa6 can inhibit MTs directly without interacting molecules via an N-terminal 18 amino acid motif (MT elimination domain/MTED) that binds tubulin and inhibits microtubule growth in vitro and cells. If N-terminal MTED-containing fragments are in the cytoplasm they abolish entire microtubule networks of mouse fibroblasts and whole axons of fly neurons. Full-length Efa6 is membrane-attached, hence primarily blocks MTs in the periphery of fibroblasts, and explorative MTs that have left axonal bundles in neurons. Accordingly, loss of Efa6 causes an increase of explorative MTs: in growth cones they enhance axon growth, in axon shafts they cause excessive branching, as well as atrophy through perturbations of MT bundles. Efa6 over-expression causes the opposite phenotypes. Taken together, our work conceptually links molecular and sub-cellular functions of cortical collapse factors to axon growth regulation and reveals new roles in axon branching and in the prevention of axonal atrophy. Furthermore, the MTED delivers a promising tool that can be used to inhibit MTs in a compartmentalised fashion when fusing it to specifically localising protein domains.

## Introduction

Axons are the cable-like neuronal extensions that wire the nervous system. They are only 0.1–15 μm in diameter (*Hoffman, 1995*), but can be up to a meter long in humans (*Debanne et al., 2011*; *Prokop, 2013*). It is a fascinating challenge to understand how axons can extend over these enormous distances and branch in orderly manners, but also how these delicate structures can be maintained for a lifetime, that is many decades in humans. It is not surprising that we gradually lose about 40% of our axons towards old age (*Calkins, 2013*; *Marner et al., 2003*), and that axon decay is a prominent neurodegenerative phenomenon (*Adalbert and Coleman, 2013*; *Fang and Bonini, 2012*; *Medana and Esiri, 2003*; *Wang et al., 2012*).

Essential for axon biology are the parallel bundles of microtubules (MTs) running all along the axon shaft; these bundles provide (1) structural support, (2) highways for life-sustaining cargo transport, and (3) a source of MTs that can leave these bundles to drive morphogenetic changes. Through being organised in this way, MTs essentially drive processes of axon growth, branching and maintenance (*Conde and Cáceres, 2009*; *Dent et al., 2011*; *Prokop, 2013*; *Voelzmann et al., 2016a*). The dynamics of MTs are orchestrated through MT-binding and -regulating proteins, for most of which we know the molecular mechanisms of function. However, such knowledge alone is usually not sufficient to explain their cellular roles.

For example, cortical collapse factors are cell surface-associated proteins which specifically inhibit MTs that approach the cell periphery. Previous reports suggested important roles for cortical collapse factors in regulating axon growth: the ARF activator Efa6 (exchange factor for ARF6) in *C. elegans* negatively impacts on developmental and regenerative axon growth (*Chen et al., 2015*; *Chen et al., 2011*; *O'Rourke et al., 2010*); the mammalian type four kinesin KIF21A also affects axon growth and links to the neurodevelopmental eye movement disorder 'congenital fibrosis of extraocular muscles' (OMIM reference #135700; *Heidary et al., 2008*; *Tiab et al., 2004*; *van der Vaart et al., 2013*). However, we can currently only hypothesise how the molecular functions of these two collapse factors link to axon growth, most likely by acting in growth cones (GCs).

GCs are the amoeboid tip structures through which axons extend to wire the nervous system during development or regeneration. The axonal MT bundles terminate in the centre of GCs; from here, single MTs splay into the actin-rich periphery of GCs. These explorative MTs can trigger extension of the entire MT bundle into their direction, thus elongating the axon (*Dent et al., 2011*; *Lowery and Vactor, 2009*; *Prokop et al., 2013*); by inhibiting such explorative MTs, cortical collapse factors could negatively impact on axon growth.

In line with this argument, and depending on where cortical collapse factors are present and functionally active, further functional predictions could be made: for example, collateral branching of axons along their shafts has been described to depend on explorative MTs that leave the parallel axonal bundles and polymerise towards the periphery (*Kalil and Dent, 2014*; *Lewis et al., 2013*; *Tymanskyj et al., 2017*; *Yu et al., 2008*). Cortical collapse factors might therefore be negative regulators of axon branching.

Other roles might concern axon maintenance: the model of 'local axon homeostasis' states that the force-enriched environment in axons biases MTs to buckle or project out of the bundle to seed pathological areas of MT disorganisation (*Hahn et al., 2019*; *Prokop, 2016*). By inhibiting off-track MTs in the axon shaft, cortical collapse factors might prevent such processes, acting in parallel to other bundle-maintaining factors. For example, spectraplakins serve as spacers that keep polymerising MTs away from the cortex by linking the tips of extending MTs to the axonal surface and guiding them into parallel bundles (*Alves-Silva et al., 2012*). Their deficiency in any organism causes severe MT disorganisation, potentially explaining human dystonin-linked HSAN6 ('type six hereditary sensory and autonomic neuropathy''; OMIM ID: 614653; *Voelzmann et al., 2017*). If our hypothesis is correct, loss of cortical collapse factors in axon shafts would also cause MT disorganisation, but through a very different mechanistic route.

Here, we make use of *Drosophila* neurons as a well-established, powerful model for studying roles of MT regulators (*He and Roblodowski, 2016*; *Nye et al., 2014*; *Prokop et al., 2013*; *Sánchez-Soriano et al., 2007*). Using in vitro and cellular assays, we show that *Drosophila* Efa6 is a cortical collapse factor acting through its N-terminal MT elimination domain (MTED). We find that the MTED binds tubulin and blocks MT polymerisation in vitro which indicates that the effect of the peptide is due to a direct interaction between the peptide and tubulin. By localising to neuronal membranes, it only abolishes explorative MTs. This subcellular role translates into negative regulation of axon growth and branching and the prevention of pathological MT disorganisation, both in cultured neurons and in vivo. We propose Efa6 to function as a quality control or axonal maintenance factor that keeps explorative MTs in check, thus playing a complementary role to spectraplakins that prevent MTs from leaving axonal bundles.

## Results

### Efa6 is widely expressed in *Drosophila* neurons and restricts axonal growth

To evaluate the function of Efa6 in neurons, we first determined its expression in the nervous system. We used a genomically engineered fly line in which the endogenous *Efa6* gene was GFP-tagged (*Efa6-GFP*; *Huang et al., 2009*). These animals widely express Efa6::GFP throughout the CNS at larval and adult stages (*Figure 1A–D*). We cultured primary neurons from this fly line to analyse the subcellular distribution of Efa6. In young neurons at 6 hr in vitro (6HIV) and in mature neurons at 5 days in vitro (5DIV), Efa6 was localised throughout cell bodies and axons (*Figure 1E,F,H,I*).

We next determined whether *Drosophila* Efa6 has an impact on axon growth, using a range of loss-of-function conditions: Efa6 knock-down (*Efa6-RNAi*), two overlapping deficiencies uncovering the entire *Efa6* gene locus (*Efa6$^{Def}$*), and three precise gene deletions including *Efa6$^{GX6[w+]}$*, *Efa6$^{GX6[w-]}$* and *Efa6$^{KO\#1}$* (see Materials and methods for details; *Huang et al., 2009*). In all these conditions, axon length at 6 HIV was increased compared to wild-type by at least 20% (*Figure 2A,B,D*). Since there were no obvious differences between the precise deletion lines, these alleles were used interchangeably for further experiments.

We then tested whether over-expression of Efa6 would cause the opposite effect, that is axon shortening or even loss. For this, we generated a transgenic *UAS-Efa6-FL-GFP* line and, in addition, developed methods to transfect *UAS*-constructs into *Drosophila* primary neurons (see Materials and methods). As similarly observed with the endogenous protein, full-length *Efa6-FL::GFP* localised to cell bodies, axons and growth cones of primary neurons also when expressed pan-neuronally using either the transgenic line (*Figure 1G*) or cell transfection (*Figure 3—figure supplement 1B*). The transgenic expression caused a ~ 20% reduction in axon length, which was increased to ~50% upon transfection (likely due to higher copy numbers of the expression construct; *Figure 2C,D*). Furthermore, we observed an increase in the number of neurons without axons

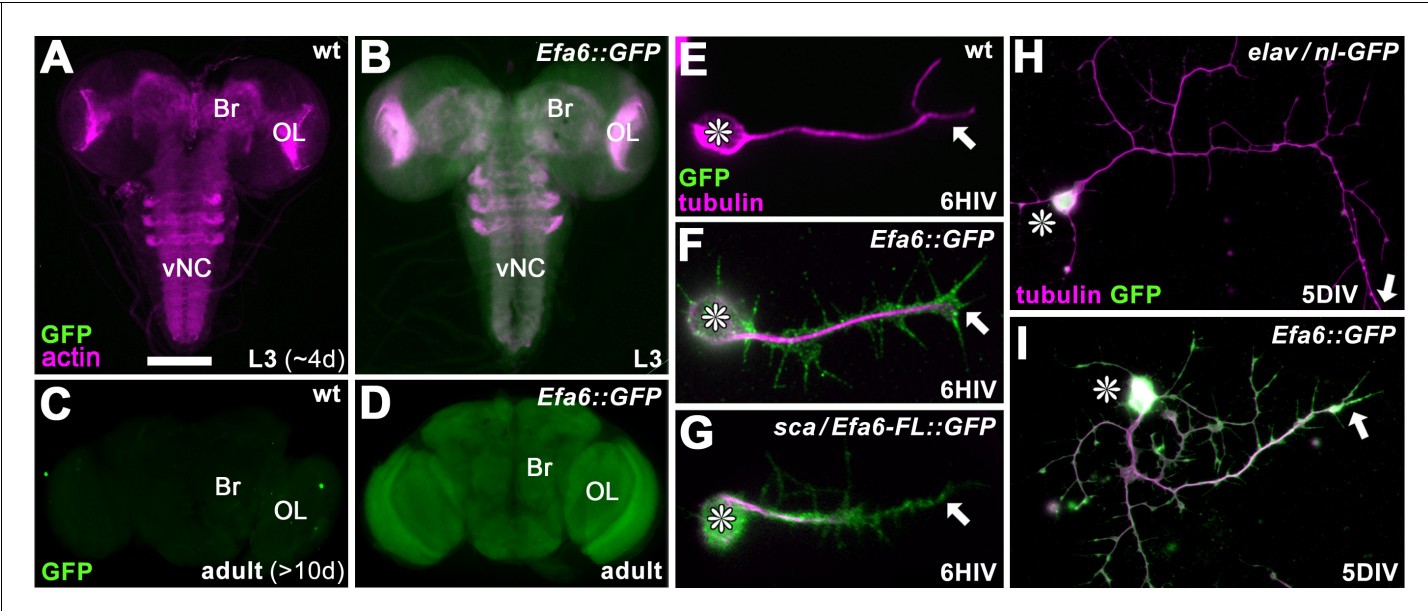

**Figure 1.** Efa6 is expressed throughout neurons at all developmental stages. (**A–B**) Late larval CNSs at about 4d after egg lay (L3; **A,B**) and adult CNSs from 10d old flies (**C,D**) derived from control wild-type animals (wt) or the Efa6::GFP line (*Efa6::GFP*), stained for GFP and actin (Phalloidin, only larval preparations); OL, optic lobe; Br, central brain; vNC, ventral nerve cord. (**E–I**) Images of primary *Drosophila* neurons at 6HIV or 5DIV (as indicated bottom right), stained for tubulin (magenta) and GFP (green); control neurons are wild-type (wt) or express *elav-Gal4*-driven nuclear GFP (*elav/nl* GFP), further neurons are either derived from the endogenously tagged Efa6::GFP line or express Efa6-FL::GFP under the control of *sca-Gal4* (*sca/Efa6-FL:: GFP*); asterisks indicate cell bodies and arrows the axon tips. Scale bar in A represents 75 μm in A and B, 130 μm in C and D, 15 μm in E-H, 25 μm in I and E.

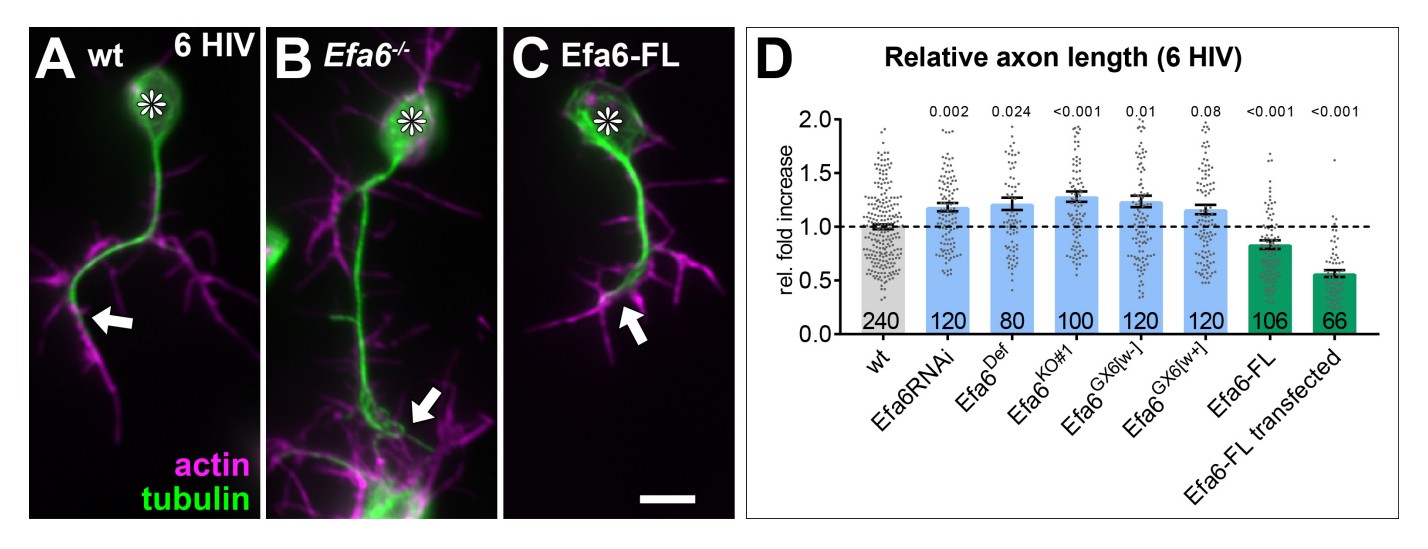

**Figure 2.** Efa6 regulates axonal length in primary *Drosophila* neurons. Examples of primary *Drosophila* neurons at 6HIV (**A–C**), all stained for actin (magenta) and tubulin (green); neurons are either wild-type controls (**A**), Efa6-deficient (**B**), or expressing Efa6-FL::GFP (**C**); asterisks indicate cell bodies, arrows point at axon tips; the scale bar in C represents 10 μm. Quantification of axon lengths at 6HIV (**D**); different genotypes are colour-coded: grey, wild-type controls; blue, different *Efa6* loss-of-function conditions; green, neurons over-expressing Efa6; data represent fold-change relative to wild-type controls (indicated as horizontal dashed 'ctrl' line); they are shown as single data points and a bar indicating mean ± SEM data; P values from Mann-Whitney tests are given above each column, sample numbers at the bottom of each bar represent individual neurons pooled from at least two replicates, that is experiments conducted on different days. For raw data see *Figure 2—source data 1*.

The online version of this article includes the following source data for figure 2:

**Source data 1.** Summary of the statistics from *Figure 2*.

from ~26% in *UAS-GFP*-transfected controls to ~43% in Efa6-FL::GFP-positive neurons (*Figure 3B,F',F''*).

Together, these results suggest that Efa6 restricts axonal growth, comparable to reports for *C. elegans* Efa6 (*Ce*Efa6; *Chen et al., 2015*; *Chen et al., 2011*). The loss of whole axons upon Efa6-FL::GFP over-expression might suggest that Efa6 performs its morphogenetic roles by inhibiting MT networks.

## Efa6 eliminates peripheral or even entire MT networks in mouse fibroblasts

To assess whether the negative impact of Efa6 on axon outgrowth might be through inhibiting MTs, we used NIH3T3 mouse fibroblasts as a heterologous cell system known to provide meaningful read-outs for functional studies of *Drosophila* MT regulators (*Alves-Silva et al., 2012*; *Beaven et al., 2015*). When fibroblasts were analysed 24 hr after transfection with *Efa6-FL-GFP*, we found a graded depletion of MT networks depending on Efa6-FL::GFP protein levels (shown and quantified in *Figure 3—figure supplement 2*). At low or moderate expression levels, Efa6-FL::GFP localised along the circumference and in areas of membrane folds (open arrow heads in *Figure 3—figure supplement 2A,B*), and MTs tended to be lost predominantly from the cell fringes (curved arrows in *Figure 3—figure supplements 2B* and *3B*). At high expression levels, Efa6-FL::GFP became detectable in the cytoplasm and even nucleus (double-chevrons in *Figure 3—figure supplement 2C*), suggesting that membrane-association might become saturated. In these cases, prominent MT networks were gone (*Figure 3—figure supplement 2C*). When quantifying these MT phenotypes across all transfected fibroblasts, there was a strong increase in MT network defects and depletion upon Efa6-FL::GFP expression, but not in GFP controls (*Figure 3B*).

When performing live analyses, we consistently observed that growing MTs labelled with EB3::mCherry extended to the very cell fringes of control fibroblasts (*Figure 4A*; *Animation 1*), whereas MTs in fibroblasts transfected with Efa6-FL::GFP showed a very different behaviour: hardly any MTs polymerised into areas along the rim where Efa6 was enriched but stopped at the border of the

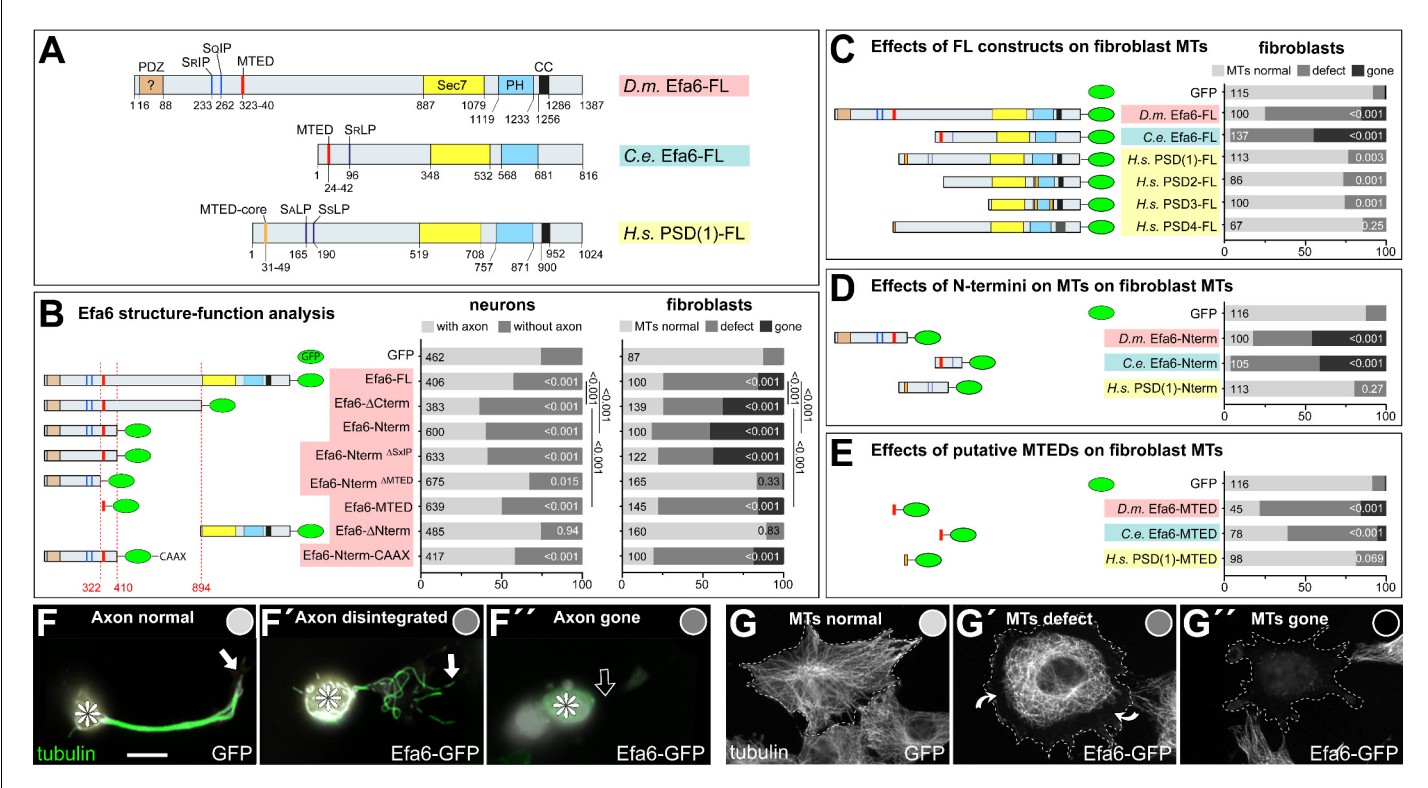

**Figure 3.** Efa6 domain and motif requirements for MT inhibition in neurons and fibroblasts. (A) Schematics of *Drosophila melanogaster (Dm)* Efa6 (isoform *C; CG31158*), *Caenorhabditis elegans (Ce)* Efa6 (isoform *Y55D9A.1a*) and *Homo sapiens (Hs)* PSD1 (isoform *201/202; NP_002770.3*), illustrating the positions (numbers indicate first and last residues) of the putative PDZ (PSD95-Dlg1-ZO1) domain [expected to anchor to transmembrane proteins (**Ponting et al., 1997**), but not mediating obvious membrane association in fibroblasts: *Figure 3—figure supplement 6C,D*], SxIP/SxLP motifs (SRIP, SQIP, SALP, SSLP), the MT elimination domain (MTED), SEC7 domain, plekstrin homology (PH) domain and coiled-coil domain (CC). (B) Schematics on the left follow the same colour code and show the *Dm*Efa6 constructs used in this study (dashed red lines indicate the last/first residue before/behind truncations). Bar graphs on the right show the impact that transfection of these constructs had on axon loss in primary *Drosophila* neurons (dark grey in left graph) and on MT loss in fibroblasts (dark grey or black as indicated; for respective images see F and G below). Analogous fibroblast experiments as performed with *Drosophila* constructs were performed with full length constructs of *C. elegans* Efa6 and human PSDs (C), with N-terminal constructs (D) or synthetic MTEDs (E) of *Dm* and *Ce*Efa6 and of human PSD1. Throughout this figure, construct names are highlighted in red for *Drosophila*, light blue for *C. elegans* and yellow for *Homo sapiens*; all graph bars indicate percentages of neurons with/without axons (light/dark grey) and of fibroblasts with normal, reduced or absent MTs (light, medium, dark grey, respectively); numbers in the left end of each bar indicate sample numbers indicating individual cells pooled from at least two replicates, on the right end the P values from Chi$^2$ tests relative to GFP controls; numbers on the right of bars in B compare some constructs to Efa6-FL::GFP, as indicated by black lines. (F–F'') Primary neurons expressing GFP or Efa6-FL::GFP transgenically and stained for tubulin (asterisks, cell bodies; white arrows, axon tips; open arrow, disintegrated or absent axon). (G–G'') Fibroblasts expressing Efa6-FL:: GFP and stained for tubulin; curved arrows indicate areas where MTs are retracted from the cell periphery; grey dots in F-G'' indicate the phenotypic categories for each neuron and fibroblasts, as used for quantitative analyses in the graphs above. Scale bar at bottom of F refers to 10 μm in F and 25 μm in G. For raw data see *Figure 3—source data 1*.

The online version of this article includes the following source data and figure supplement(s) for figure 3:

**Source data 1.** Summary of the statistics from *Figure 3*.
**Figure supplement 1.** Localisation of Efa6 constructs in primary neurons.
**Figure supplement 2.** MT inhibition by Efa6-FL is concentration-dependent in fibroblasts.
**Figure supplement 2—source data 1.** Summary of the statistics from *Figure 3—figure supplement 2*.
**Figure supplement 3.** Representative MT phenotypes induced by the different constructs in transfected fibroblasts.
**Figure supplement 4.** Efa6 N-terminal domains vary amongst different phyla.
**Figure supplement 5.** Phylogenetic tree analysis of Efa6.
**Figure supplement 6.** Efa6 constructs localisations in fibroblasts.
**Figure supplement 7.** Conserved functions of the Efa6 C-terminus in membrane ruffle formation.
**Figure supplement 7—source data 1.** Summary of the statistics from *Figure 3—figure supplement 7*.

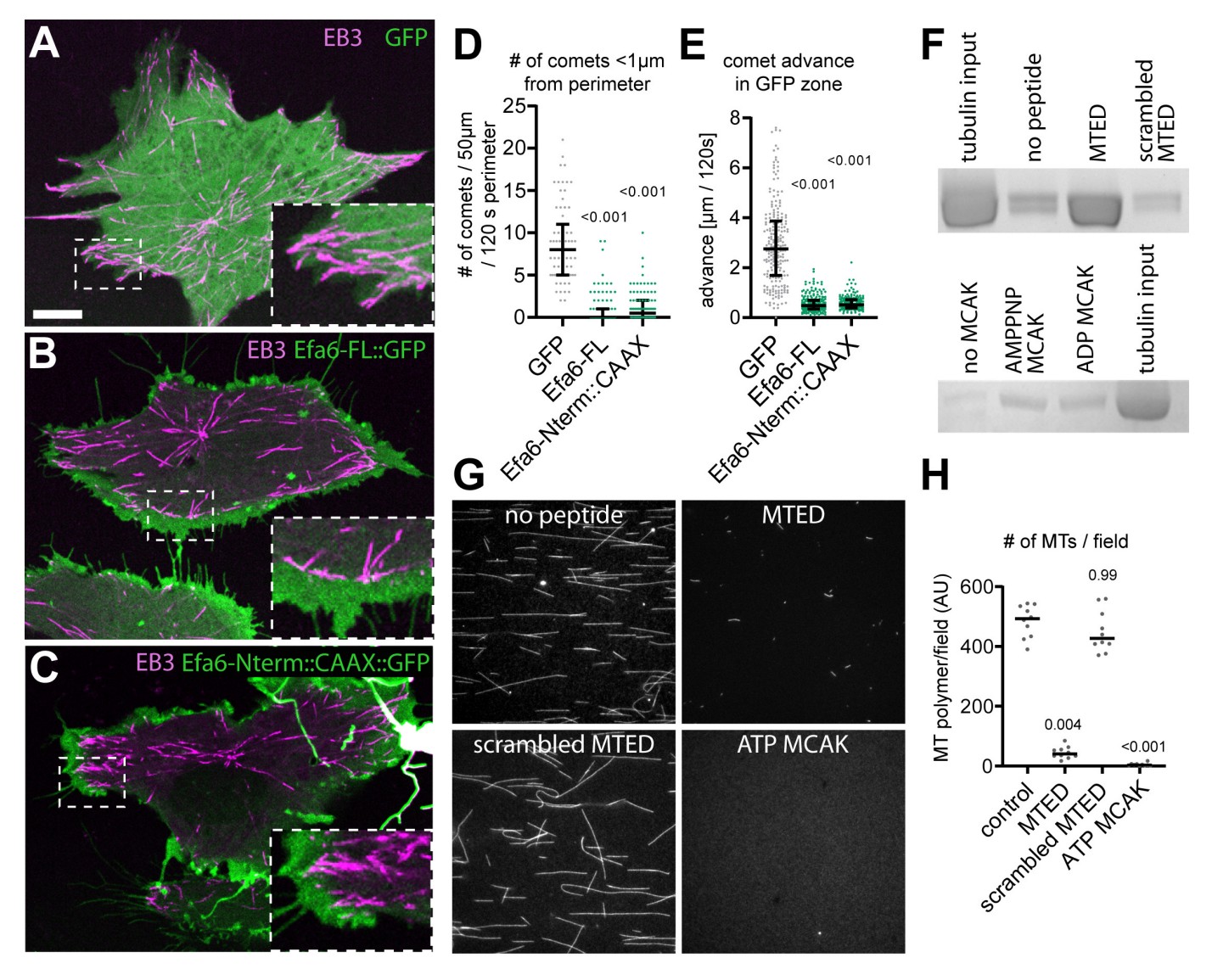

**Figure 4.** EFA6 peptide interacts directly with α/β-tubulin and inhibits microtubule growth. (**A–C**) Fibroblasts co-expressing Eb3::RFP (magenta) together with either GFP (**A**), Efa6-FL::GFP (**B**) or Efa6-Nterm::GFP::CAAX (**C**; all shown in green); images are maximum intensity projections of all frames taken at 1 s intervals during a 120 s live imaging period; stippled areas are shown as 2.5fold magnified insets. Example movies are provided as *Animations 1*, *2* and *4*. (**D,E**) Quantification of Eb3::RFP comet behaviours deduced from images comparable to those shown in A to C: in D, each data point represents the number of comets that were within a 1 μm range from the perimeter assessed over a 50 μm perimeter stretch, respectively (n = 73 for GFP, 90 for Efa6-FL::GFP and 90 for Efa6-Nterm::GFP::CAAX); in E, each data point represents the distance individual Eb3 comets reached into GFP-positive areas (in B and C assessed in areas of high GFP expression, in A in areas close to the perimeter; n = 224 for GFP, 238 for Efa6-FL::GFP and 197 for Efa6-Nterm::GFP::CAAX). (**F**) Pull-down of porcine brain tubulin using sepharose beads which were either uncoated, or coated with MTED, with a scrambled version of MTED or with MCAK (in the presence of ADP or the ATP analogue AMPPNP); proteins/peptides were randomly attached via cyanogen bromide coupling of lysines and/or N-terminal amines; quantification data are provided as *Figure 4—source data 1*. (**G**) Fluorescence images of rhodamine-labelled MTs grown in the presence of no peptide, MTED, scrambled MTED or MCAK together with ATP. (**H**) Quantification of the amount of MT polymer per field of view (n = 10 fields in each case; AU, arbitrary unit). Numbers above plots in D, E and F represent P-values determined via Kruskal–Wallis one-way ANOVA with *post hoc* Dunn's test. Scale bar in A represents 10 μm in A-C and three in G. For raw data see *Figure 4—source data 1*.

The online version of this article includes the following source data and figure supplement(s) for figure 4:

**Source data 1.** Summary of the statistics from *Figure 4*.

**Figure supplement 1.** In vitro attempts to resolve the MT inhibition mechanism of Efa6.

**Figure supplement 1—source data 1.** Summary of the statistics from *Figure 4—figure supplement 1*.

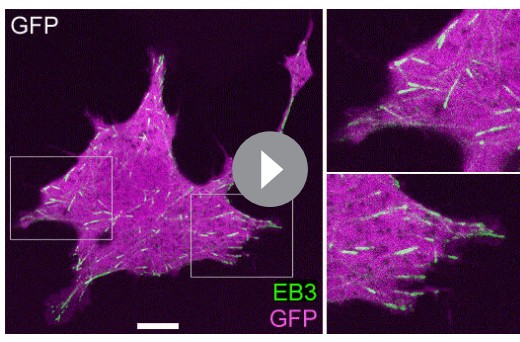

**Animation 1.** MT behaviours in control fibroblasts expressing GFP. Transfected fibroblasts co-expressing EB3::RFP (green) to visualise MT polymerisation, together with GFP (magenta) which localises throughout the cell. White boxed areas are shown as twofold magnified close-ups on the right. MTs clearly polymerise to the very perimeter of the transfected cell (quantifications are shown in *Figure 4*). Movies were taken at one frame per second. The scale bar corresponds to 25 µm in the main image.

https://elifesciences.org/articles/50319#video1

expression zone, often accompanied by Efa6-FL:: GFP accumulation at at the invasion sites of MT plus ends (*Figure 4B,D,E*; *Animation 2*).

Taken together, these fibroblast experiments confirm MT-inhibiting functions of Efa6. They suggest that Efa6 is membrane-associated and excludes MTs from this position, which may similarly apply to roles of Efa6 in neuronal morphogenesis. We concluded that the complementary use of mouse fibroblasts and *Drosophila* primary neurons provides an informative combination of readouts for MT loss and axon morphology - ideal to carry out a systematic structure-function analysis of Efa6.

## The N-terminal 18aa motif of Efa6 is essential for microtubule-inhibiting activity of Efa6

A detailed analysis of the domain structures of Efa6 proteins from 30 species revealed that C-termini of almost all species contain a putative pleckstrin homology domain (PH; potentially membrane-associating; *Macia et al., 2008*), a Sec7 domain (potentially activating Arf GTPases; *D'Souza-Schorey and Chavrier, 2006*; *Huang et al., 2009*) and a coiled-coil domain (CC; *Franco et al., 1999*; *Figure 3A*, *Figure 3—figure supplement 4A*). In contrast, the N-termini are mainly unstructured and reveal enormous length differences among species. Accordingly, phylogenetic relationship analyses comparing either full-length or N-terminal Efa6, show that chordate proteins are rather distant from invertebrates, and that arthropods form a clear subgroup within the invertebrates (*Figure 3—figure supplement 5A,B*). None of the identifiable N-terminal domains/ motifs is particularly well conserved (details in *Figure 3A*, *Figure 3—figure supplement 4A*). For example, the *Drosophila* N-terminus contains (1) a putative PDZ domain (aa16-88; mainly found in insect versions of Efa6), (2) two SxIP motifs (aa 233–6 and 262–5; found primarily in Efa6 of flies, some other insects and molluscs; some vertebrate/mammalian species display derived SxLP motifs), and (3) a motif of 18aa (from now on referred to as MT elimination domain, MTED) displaying 89% similarity with a motif in the N-terminus of *Ce*Efa6 suggested to be involved in MT inhibition (*O'Rourke et al., 2010*; conserved in nematodes, arthropods and molluscs).

To assess potential roles of the 18aa long *Drosophila* MTED, we generated a series of GFP-tagged N-terminal constructs (*Figure 3B*): *Efa6-ΔCterm-GFP* (encoding the entire N-terminal half upstream of the Sec7 domain), *Efa6-Nterm-GFP* (restricting to the N-terminal part containing all the identified functional domains), *Efa6-Nterm^{ΔMTED}-GFP* (lacking the MTED) and *Efa6-MTED-GFP* (encoding only the MTED). All these N-terminal Efa6 variants showed localisation throughout neurons (*Figure 3—figure*

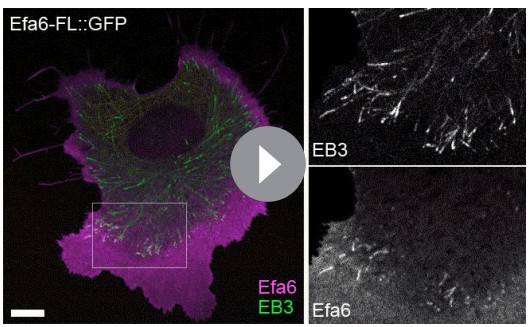

**Animation 2.** MT behaviours in control fibroblasts expressing Efa6-FL::GFP. Transfected fibroblasts co-expressing EB3::RFP (green) to visualise MT polymerisation, together with Efa6-FL::GFP (magenta) which localises primarily to the cell periphery. White boxed area is shown as twofold magnified close-up on the right, separated by channel. MTs clearly fail to polymerise into the Efa6-FL::GFP-enriched zone and hardly ever reach the cell perimeter (quantifications are shown in *Figure 4*). Movies were taken at one frame per second. The scale bar corresponds to 25 µm in the main image.

https://elifesciences.org/articles/50319#video2

supplement 1C,D,F,G), and in the cytoplasm and nucleus of fibroblasts (*Figure 3—figure supplement 6C,D,F,G*). Cytoplasmic and nuclear localisations occurred even at low expression levels, indicating that the absent C-terminus (and likely PH domain within) usually mediates membrane association. This nuclear localisation occurs in the absence of any predicted N-terminal nuclear localisation sequences (*Figure 3A*, *Figure 3—figure supplement 4A*), likely reflecting a known artefact of GFP-tagged proteins (*Alves-Silva et al., 2012*; *Seibel et al., 2007*).

In spite of their very similar localisation patterns, the functional impact of these constructs was clearly MTED-dependent: only constructs containing the MTED (Efa6-ΔCterm::GFP, Efa6-Nterm:: GFP and Efa6-MTED::GFP) caused strong axon loss in neurons and MT network depletion in fibroblasts, whereas Efa6-Nterm$^{\Delta MTED}$::GFP caused no phenotypes in fibroblasts and very mild axon loss in neurons (*Figure 3B*; *Figure 3—figure supplement 1C,D,F,G*; *Figure 3—figure supplement 3C, D,G,H*). Potential tendencies of Efa6-Nterm$^{\Delta MTED}$::GFP to cause some aberration, were complemented by findings that MTED::GFP-induced effects were less strong than those by two longer N-terminal constructs (*Figure 3B*). This might suggest that there are some relevant N-terminal motifs or regions outside the MTED.

Likely candidates are the two SxIP motifs predicted to bind EB proteins (*Honnappa et al., 2009*; *Figure 3A*). Accordingly, we found that Efa6-Nterm::GFP tip-tracks, and that Efa6-FL::GFP and EB3:: RFP co-localise at points where MTs enter Efa6-FL::GFP-rich areas in fibroblast (*Animations 2* and *3*). Such binding to EBs at MT plus ends, might enhance Efa6's ability to capture MTs for inhibition. However, when replacing each of the two SxIP motifs by four alanines, we found that the resulting Efa6-Nterm$^{\Delta SxIP}$::GFP construct induced similarly strong axon loss in neurons and MT network depletion in fibroblasts as observed with Efa6-Nterm::GFP (*Figure 3B*, *Figure 3—figure supplement 1D, E*, *Figure 3—figure supplement 3D,F*, *Figure 3—figure supplement 6D,E*). Similar observations were reported for Kif2C which clearly tip-tracks MTs in an EB1-dependent manner but does not require this property for its MT-depolymerising activity (*Moore et al., 2005*).

In conclusion, there might be some unidentified support sequences in the N-terminus, but our results clearly pinpoint the MTED as the key mediator of *Drosophila* Efa6's MT-depleting functions, suggesting this function to be conserved between flies and *C. elegans*.

## The MTED is a good predictor of MT-inhibiting function directly affecting MT polymerisation

To assess whether the MTED motif is a good predictor for MT-inhibiting capabilities of Efa6 family members, we used 12 different constructs: full length versions of (1) *Ce*Efa6, (2) *Drosophila* Efa6 and (3-6) all four human PSDs (*Figure 3C*), as well as N-terminal versions of (7) *Ce*Efa6, (8) fly Efa6 and (9) human PSD1 (*Figure 3D*). Furthermore, we deduced a MTED consensus sequence from 39 *Efa6* genes (details in *Figure 3—figure supplement 4C*), identified the most likely human MTED-like sequence (position 31-49aa of PSD1; MTED-core in *Figure 3A*) and synthesised codon-optimised versions of (10) this human as well as the (11) fly and (12) worm MTEDs (*Figure 3—figure supplement 4B,C*). When transfected into fibroblasts, we found that all six fly/worm constructs had strong MT-inhibiting properties, whereas the six human constructs (PSD1-4 full length, PSD1-Nterm, PSD1-MTED-like) showed only a slight increase in MT network defects that were far from the strong MT depletion observed with the fly/worm constructs; complete depletion of MT networks was never observed with the human constructs (*Figure 3C– E*). Therefore, the presence of a well-conserved

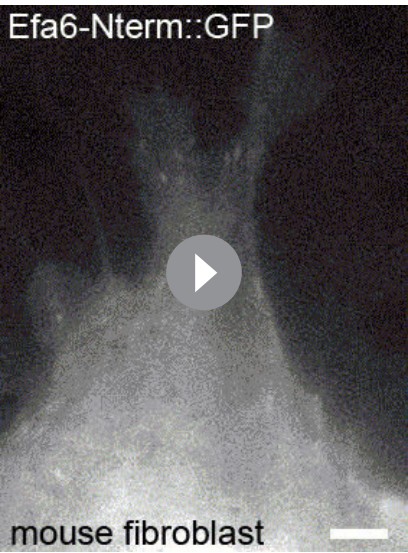

**Animation 3.** Efa6-Nterm::GFP tip-tracks in fibroblasts. Upon Efa6-Nterm::GFP expression in mouse NIH3T3 fibroblast about 6 hr after transfection, comets were observed in a few instances. Pictures for the movie were taken at 2 s intervals. Scale bar indicates 10 μm. https://elifesciences.org/articles/50319#video3

canonical MTED seems to be a good predictor for MT-inhibiting capabilities of Efa6 proteins.

To gain insights into the mechanisms through which MTEDs might act, we carried out a series of in vitro experiments. Purified Efa6-Nterm::GFP clearly associated with MTs in vitro (*Figure 4—figure supplement 1A*), but failed to depolymerise MTs (*Figure 4—figure supplement 1B*). We therefore tested the same protein in *Xenopus* egg extract to assess potential co-factor requirements, but saw again no activity (*Figure 4—figure supplement 1C*) - in spite of the fact that injection of a corresponding mRNA into *Xenopus* eggs caused strong cell division phenotypes (*Figure 4—figure supplement 1D,E*).

We suspected problems with recombinant expression of Efa6-Nterm::GFP which is predicted to have large disordered regions. We used therefore synthetic MTED peptide as an alternative approach and tested it in MT growth assays (*Figure 4G,H*). We found that the amount of MT polymer produced in control MT growth assay conditions without peptide, was approximately 10-fold reduced when synthetic MTED peptide was added, whereas a scrambled version of the peptide failed to show this effect (*Figure 4G,H*). The effect of the MTED peptide is not as potent as that of the well-characterised MT depolymerising kinesin, KIF2C/MCAK (mitotic centromere-associated kinesin), which can completely inhibit MT growth at lower concentrations but requires ATP to this end (*Figure 4G,H*).

The MT growth assay suggested that MTED can directly interfere with MT polymerisation in the absence of any auxiliary factors, and the decoration of MTs with Efa6-Nterm::GFP in our in vitro binding assays (*Figure 4—figure supplement 1A*) is in agreement with this notion. To confirm direct interaction with tubulin, we coated sepharose beads with MTED and pulled down un-polymerised GDP-tubulin. We found that the MTED-coated beads pulled down three times more tubulin than the same beads uncoated, and five times more tubulin than beads coated with a scrambled version of the peptide (*Figure 4F*, top). These data indicate a direct interaction of the MTED peptide with tubulin and suggest that a scrambled version of this peptide appears to passivate the bead surface reducing non-specific interaction of tubulin. When using the same set-up for MCAK and comparing it to uncoated beads, approximately 1.5 times more tubulin is pulled down by beads coated with MCAK in the presence of either AMPPNP or ADP (*Figure 4F* bottom; MCAK is known to strongly bind tubulin in the AMPPNP state and weakly bind tubulin in the ADP state; *Wagenbach et al., 2008*). In these experiments, the overall amount of tubulin pulled down with MCAK was considerably lower than with MTED; this is not surprising when considering (1) that far fewer of the large MCAK molecules can be accommodated on the beads, (2) that the tubulin-binding domain via its four lysines could be a frequent site of bead attachment, thus blocking access for tubulin, and (3) that the conditions required for this method of bead attachment were less favourable for retention of activity than in previous binding assays (*Wagenbach et al., 2008*).

Taken together, the MTED exists primarily in Efa6 homologues of invertebrate species and its presence correlates with MT-inhibiting properties of these proteins. This conclusion is strongly supported by our finding that *Drosophila* MTED directly interferes with MT polymerisation. This can explain why MTs fail to enter Efa6-enriched areas in fibroblasts (*Figure 4B*; *Animation 2*).

## The C-terminal domain restricts the microtubule-inhibiting activity of Efa6 to the cortex

Our structure-function analyses strongly suggested that Efa6 is membrane-associated. This is further supported by the observation that fibroblasts expressing Efa6-FL::GFP or the C-terminal derivative Efa6-ΔNterm::GFP (*Figure 3B*), display a membrane ruffle phenotype (curved open arrows in *Figure 3—figure supplements 6B,H* and *7B,D*). Efa6-ΔNterm::GFP had no obvious effects on MT networks (*Figure 3—figure supplement 3I*), and its membrane ruffling phenotype likely reflects an evolutionarily conserved function of the Efa6 C-terminus through its Sec7, PH and/or CC domains (*Derrien et al., 2002*; *Franco et al., 1999*; *Macia et al., 2008*). Accordingly, we find the same membrane ruffling when expressing PSD1-FL::GFP (curved open arrows in *Figure 3—figure supplement 7E*).

However, even if the C-terminus plays no active role in the MT inhibition process, it still regulates this function: the Efa6-ΔCterm::GFP and Efa6-Nterm::GFP variants which lack the C-terminus (*Figure 3B*), fail to associate with the cortex (*Figure 3—figure supplement 6C,D*), do not cause ruffling (*Figure 3—figure supplement 7C*), but induce MT phenotypes far stronger than Efa6-FL::GFP does (*Figure 3B* and *Figure 3—figure supplement 3C,D,H vs* B). To assess whether lack of

membrane tethering could explain this phenotypic difference, we generated the Efa6-Nterm::GFP::CAAX variant (*Figure 3B*) where Efa6-Nterm::GFP is fused to the membrane-associating CAAX domain (*Hancock et al., 1991*); this addition of CAAX changed the properties of Efa6-Nterm::GFP back to Efa6-FL::GFP-like behaviours: the hybrid protein localised to the cortex and caused only a moderate MT phenotype in fibroblasts, the axon loss phenotype was mild (*Figure 3B*; *Figure 3—figure supplement 6B,I*; 3B,E) and, also in live analyses, the CAAX construct reproduced the effect of excluding MTs from Efa6-N-term::GFP::CAAX-enriched areas, confirming that this inhibition is mediated by the N-terminus (*Figure 4C–E*; *Animation 4*). These findings confirm membrane tethering as an important regulatory feature restricting Efa6 function.

Taken together, our structure-function data clearly establish *Drosophila* Efa6 as a cortical collapse factor: its N-terminal MTED blocks MT polymerisation, and this function is restricted to the cortex through the Efa6 C-terminus which associates with the cell membrane.

## Efa6 negatively regulates MT polymerisation at the growth cone membrane and in filopodia

We next asked how Efa6's cortical collapse function relates to the observed axon growth phenotypes. For this, we focussed on growth cones (GCs) as the sites where axons extend; this extension requires the splaying of MTs from the axonal bundle tip at the base of GCs to explore the actin-rich periphery (*Dent et al., 2011*; *Lowery and Vactor, 2009*; *Prokop et al., 2013*).

In GCs of primary neurons at 6 HIV, loss of Efa6 caused an increase in MT polymerisation events: the total number of Eb1 comets was increased as compared to wild-type controls (*Figure 5I*). Eb1::GFP comets in wild-type neurons tended to die down upon hitting the plasma membrane within GCs or at the tip of their filopodia, whereas they frequently persisted in *Efa6* mutant neurons, where they could occasionally even be observed to undergo curved extensions along the periphery (*Animations 5–8*). Comet velocity was unaffected (~0.14 µm/s), but the lifetime of Eb1::GFP comets (5.04 s ± 0.60 SEM in wild-type) was ~1.4 times longer in mutant GCs, with the dwell time of Eb1 comets at the tip of filopodia being ~3 fold increased (from 2.10 s ± 0.24 SEM in wild-type to 6.26 s± 0.40 SEM in *Efa6* mutant neurons; *Figure 5M*); we even observed cases where comets at the tips of filopodia were moving backwards, seemingly pushed back by the retracting filopodial tips (*Animations 7* and *8*). In agreement with the increased lifetime, more MTs were found in growth cone filopodia in *Efa6* mutant neurons, as quantified by counting filopodia that contained EB1 comets or MTs (*Figure 5J,K* and arrow heads in D,E; note that the total number of filopodia per GC was in the range of 10–11 for both wild-type and *Efa6*; not shown). Transgenically expressed Efa6-FL::GFP caused the opposite effect, that is a reduction in the number of GC filopodia containing Eb1 comets or MTs (*Figure 5G–L*; green columns in I-L).

Next, we investigated MT dynamics in axon shafts. In contrast to GCs, MTs in the axon shaft are organised into bundles, hence kept away from the membrane. Accordingly, neither loss- nor gain-of-function had an obvious effect on Eb1 comet numbers, lifetimes, velocities and directionalities (*Figure 6A–D*). However, like in GCs, there was a strong increase in filopodia along the shaft that contained MTs when Efa6 was absent, and a strong decrease when over-expressing Efa6-FL::GFP (*Figure 6E–G*).

Taken together, our data are consistent with a model in which Efa6 primarily inhibits explorative MTs that leave the axon bundle in either GCs or axon shafts and polymerise towards the

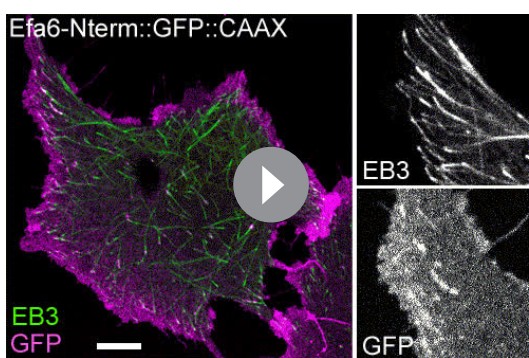

**Animation 4.** MT behaviours in control fibroblasts expressing Efa6-Nterm::GFP::CAAX. Transfected fibroblasts co-expressing EB3::RFP (green) to visualise MT polymerisation, together with Efa6-Nterm::GFP::CAAX (magenta) which localises primarily to the cell periphery. The close-up on the right is two-fold magnified and refers to the top left corner of the main image; it is separated by channel. MTs fail to polymerise into the Efa6-Nterm::GFP::CAAX-enriched zone and hardly ever reach the cell perimeter (quantifications are shown in *Figure 4D,E*). Movies were taken at one frame per second. The scale bar corresponds to 25 µm in the main image.
https://elifesciences.org/articles/50319#video4

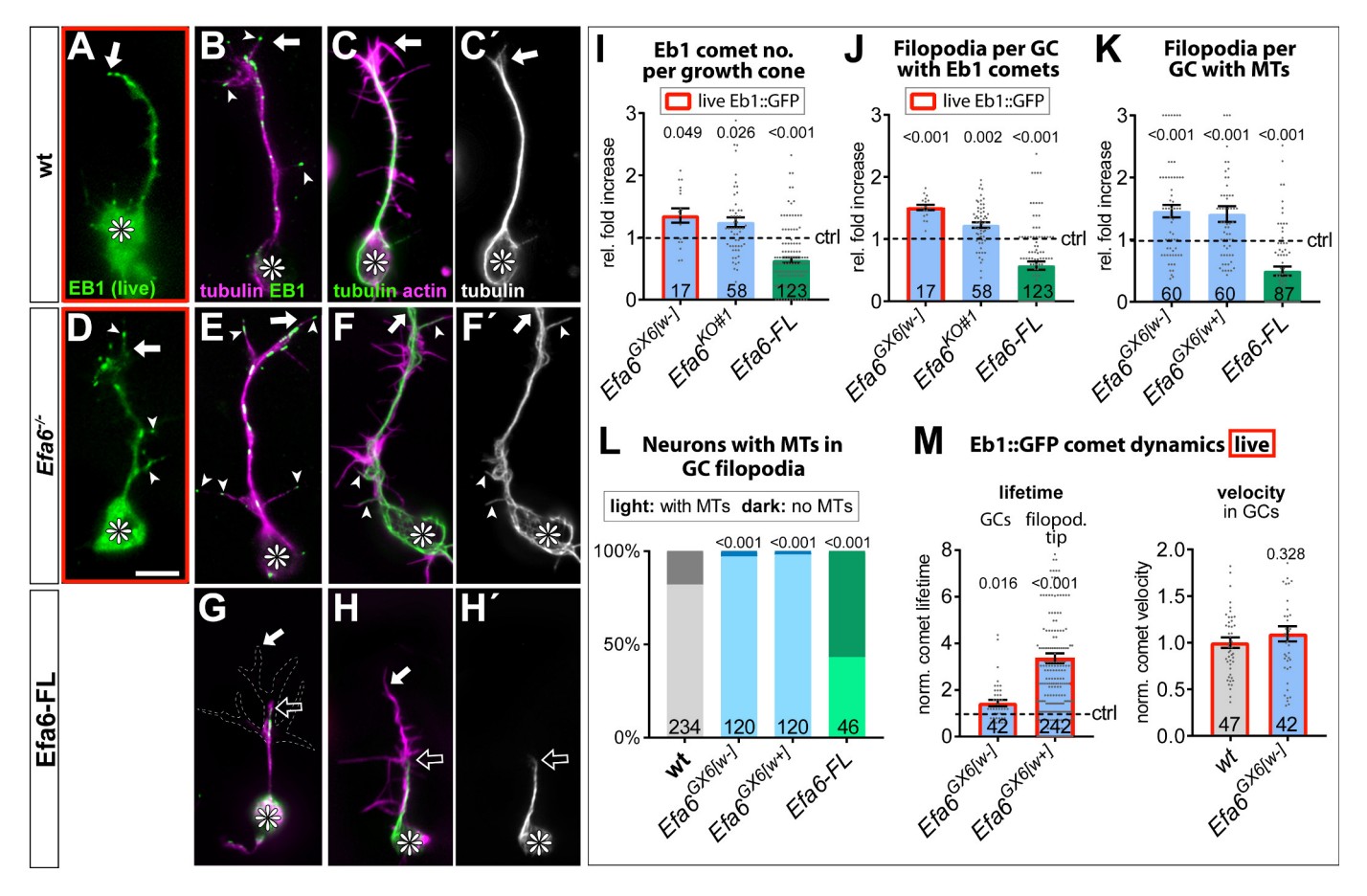

**Figure 5.** Efa6 regulates MT behaviours in GCs. (A–H') Examples of primary neurons at 6HIV which are either wild-type controls (top), Efa6-deficient (middle) or expressing Efa6-FL::GFP (bottom); neurons were either imaged live for Eb1::GFP (green in A,D) or fixed and labelled for Eb1 and tubulin (B, E,G; as colour-coded) or actin and tubulin (C,F,H; as colour coded; tubulin shown as single channel image on the right); asterisks indicate cell bodies, white arrows the tips of GCs, open arrows the tips of MT bundles, arrow heads filopodial processes containing MTs or Eb1 comets; the GC in G is outlined with a white dashed line; scale bar in D represents 5 µm in all images. (I–M) Quantitative analyses of MT behaviours in GCs, as indicated above each graph. Different genotypes are colour-coded: grey, wild-type controls; blue, different *Efa6* loss-of-function conditions; green, neurons over-expressing Efa-FL::GFP. The graph in L shows percentages of neurons without any MTs in shaft filopodia (dark shade) versus neurons with MTs in at least one filopodium (light shade; P values above bars assessed via Chi$^2$ tests), whereas all other graphs show single data points and a bar indicating mean ± SEM, all representing fold-increase relative to wild-type controls (indicated as horizontal dashed 'ctrl' line; P values above columns are from Mann-Whitney tests). The control values in M (dashed line) equate to an Eb1 comet life-time of 2.10 s ± 0.24 SEM in filpodia and 5.04 s ± 0.60 SEM in growth cones, and a comet velocity of 0.136 µm/s ± 0.01 SEM. Throughout the figure, sample numbers are shown at the bottom of each bar where each data point represents one GC (I–K), one neuron (L) or one Eb1::GFP comet (M), respectively; in all cases data were pooled from at least two replicates; data obtained from live analyses with Eb1::GFP are framed in red. For raw data see *Figure 5—source data 1*.

The online version of this article includes the following source data for figure 5:

**Source data 1.** Summary of the statistics from *Figure 5*.

cell membrane or into filopodia. Surplus MTs in the periphery of GCs can explain the extra axonal growth we observed (*Figure 2*).

## Efa6 negatively influences axon branching

We hypothesised that an increase in explorative MTs could also cause a rise in axon branching (see Introduction), either by inducing GC splitting through parallel growth events in the same GC (*Acebes and Ferrús, 2000*), or by seeding new collateral branches along the axon shaft (*Kalil and Dent, 2014*; *Lewis et al., 2013*). To test this possibility, we studied mature primary neurons at 5 days in vitro (DIV). We found that *Efa6*$^{KO\#1}$ homozygous mutant neurons showed almost double the

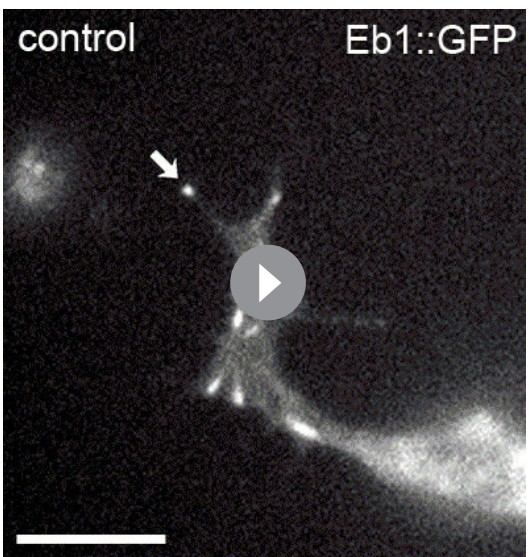

**Animation 5.** Eb1::GFP in a growth cone of a wild-type *Drosophila* primary neuron at 6 HIV. Arrows indicate positions where individual Eb1::GFP comets terminate. Pictures of the movie were taken at 2 s intervals. Scale bar indicates 10 µm.

https://elifesciences.org/articles/50319#video5

number of collateral branches as observed in wild-type neurons, whereas expression of Efa6-FL::GFP reduced branching by 21% (*Figure 7A–C,E*). This reduction is mediated by the Efa6 N-terminus, since expression of Efa6-Nterm::GFP::CAAX caused a similar degree in branch reduction (*Figure 7D,E*).

To extend these studies to neurons in vivo, we studied dorsal cluster neurons, a subset of neurons with stereotypic axonal projections in the optic lobe of adult brains (*Hassan et al., 2000*; *Voelzmann et al., 2016b*). To manipulate Efa6 levels in these neurons, either *Efa6-RNAi* or *Efa6-FL-GFP* was co-expressed with the membrane marker *myr-tdTomato*. Brains of young and old flies (2–5 d and 15–18 d after eclosure from the pupal case, respectively) were assessed with anti-GFP for specific Efa6-FL::GFP expression (*Figure 7—figure supplement 1*), and tdTomato was used to visualise axonal morphology (*Figure 7F–K*). We found that *Efa6* knock-down in dorsal cluster neurons caused a significant increase in branch numbers by 29% in young and by 38% in old brains, whereas over-expression of Efa6::GFP strongly decreased branch numbers by 33% in young and 28% in old brains, respectively (*Figure 7L*).

In these experiments, Efa6-FL::GFP expression had an intriguing further effect: Only 57% of young brains had any axons in the medulla region, compared to 88% in controls, whereas the axons were eventually present in the older Efa6-FL::GFP expressing fly brains (*Figure 7H,K,M*, *Figure 7—figure supplement 1B,D*). This suggests a delayed outgrowth phenotype, as would be consistent with decreased axon growth observed upon

**Animation 6.** Eb1::GFP in a growth cone of a wild-type *Drosophila* primary neuron at 6 HIV. Arrows indicate positions where individual Eb1::GFP comets terminate. Pictures of the movie were taken at 2 s intervals. Scale bar indicates 10 µm.

https://elifesciences.org/articles/50319#video6

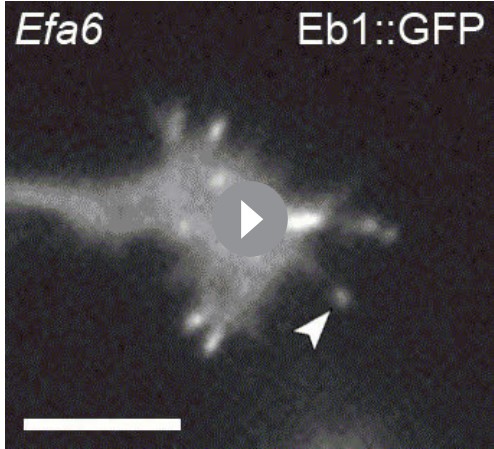

**Animation 7.** Eb1::GFP in a growth cone of an *Efa6*[GX6] [w-] mutant *Drosophila* primary neuron at 6 HIV. The arrow heads follow individual Eb1::GFP comets illustrating either their trajectories adjacent to the membrane or prolonged dwell time at filopodial tips. Pictures for the movie were taken at 2 s intervals. Scale bar indicates 10 µm.

https://elifesciences.org/articles/50319#video7

Efa6 over-expression in primary neurons (green bars in *Figure 2D*).

Taken together, our data indicate a physiologically relevant role for Efa6 as negative regulator of axonal branching, mediated through its N-terminus, most likely via its function as cortical collapse factor.

### *Efa6* maintains axonal MT bundle integrity in cultured neurons

Apart from changes in growth and branching, we noticed that a significant amount of Efa6-depleted neurons displayed axons with swellings where MTs lost their bundled conformation and were arranged into intertwined, criss-crossing curls instead (arrowheads in *Figure 8D–F*). To quantify the strength of this phenotype, we measured the area of MT disorganisation relative to axon length (referred to as' MT disorganisation index', MDI; *Qu et al., 2017*). MDI measurements in *Efa6* mutant neurons revealed a mild 1.3 fold increase in MT disorganisation in young neurons which gradually worsened to 2.3 fold at 5 DIV and ~4 fold at 10 DIV (*Figure 8A–F,I*; all normalised to controls).

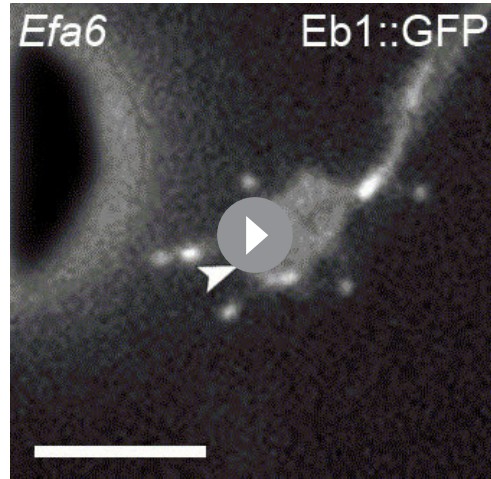

**Animation 8.** Eb1::GFP in a growth cone of an *Efa6* [GX6] [w-] mutant *Drosophila* primary neuron at 6 HIV. The arrow heads follow individual Eb1::GFP comets illustrating either their trajectories adjacent to the membrane or prolonged dwell time at filopodial tips. Pictures for the movie were taken at 2 s intervals. Scale bar indicates 10 μm.
https://elifesciences.org/articles/50319#video8

The observed gradual increase in phenotype could be the result of a genuine function of *Efa6* not only during axon growth but also their subsequent maintenance. Alternatively, it could be caused by maternal gene product deposited in the mutant embryos by their heterozygous mothers (*Roote and Prokop, 2017*); such maternal *Efa6* could mask mutant phenotypes at early stages so that they become apparent only after most Efa6 has degraded. To assess the latter possibility, we used a pre-culture strategy to remove potential maternal Efa6 (see Materials and methods for details; *Prokop et al., 2012*; *Sánchez-Soriano et al., 2010*). When plating neurons after 5 days of pre-culture, we still found a low amount of MT disorganisation in young neurons and a subsequent gradual increase to severe phenotypes over the following days (*Figure 8J*).

This finding argues for a continued role of Efa6 in preventing MT disorganisation during development as well as in mature neurons. To further test this possibility, we used a temperature-based conditional knock-down technique (*elav-GAL4 UAS-Efa6-RNAi UAS-Gal80$^{ts}$* abbreviated to *elav/Efa6$^{IR}$/Gal80$^{ts}$*; see Materials and methods for details): the *elav/Efa6$^{IR}$/Gal80$^{ts}$* neurons were grown without knock-down (19°C) for 3 days, a stage at which they have long undergone synaptic differentiation (*Küppers-Munther et al., 2004*); at that point, we found no difference in MT disorganisation between non-induced construct-bearing cells and control neurons (*Figure 8K*). After this period, cells were grown for another four days under knock-down conditions (27°C), and then fixed on day seven. At this point, MT disorganisation in the *elav/Efa6$^{IR}$/Gal80$^{ts}$* neurons was significantly increased over control neurons (*Figure 8K*), indicating that Efa6 is not only required during development but also during later maintenance to prevent MT disorganisation.

In contrast to increased MT disorganisation upon functional loss of Efa6, expression of Efa6-FL::GFP or Efa6-Nterm::GFP::CAAX showed a tendency to reduce MT disorganisation even below the baseline levels measured in control cells (cultured in parallel without the expression construct; *Figure 8I*), suggesting that also this role of Efa6 is likely due to the cortical collapse function of Efa6 (see Discussion).

### *Efa6* maintains axonal MT bundle integrity in vivo

We then assessed whether a role of Efa6 in MT bundle maintenance is relevant in vivo. For this, we studied a subset of lamina neurons, which project prominent axons in the medulla of the adult optic lobe (*Prokop and Meinertzhagen, 2006*). We labelled MTs in these axons by expressing α-

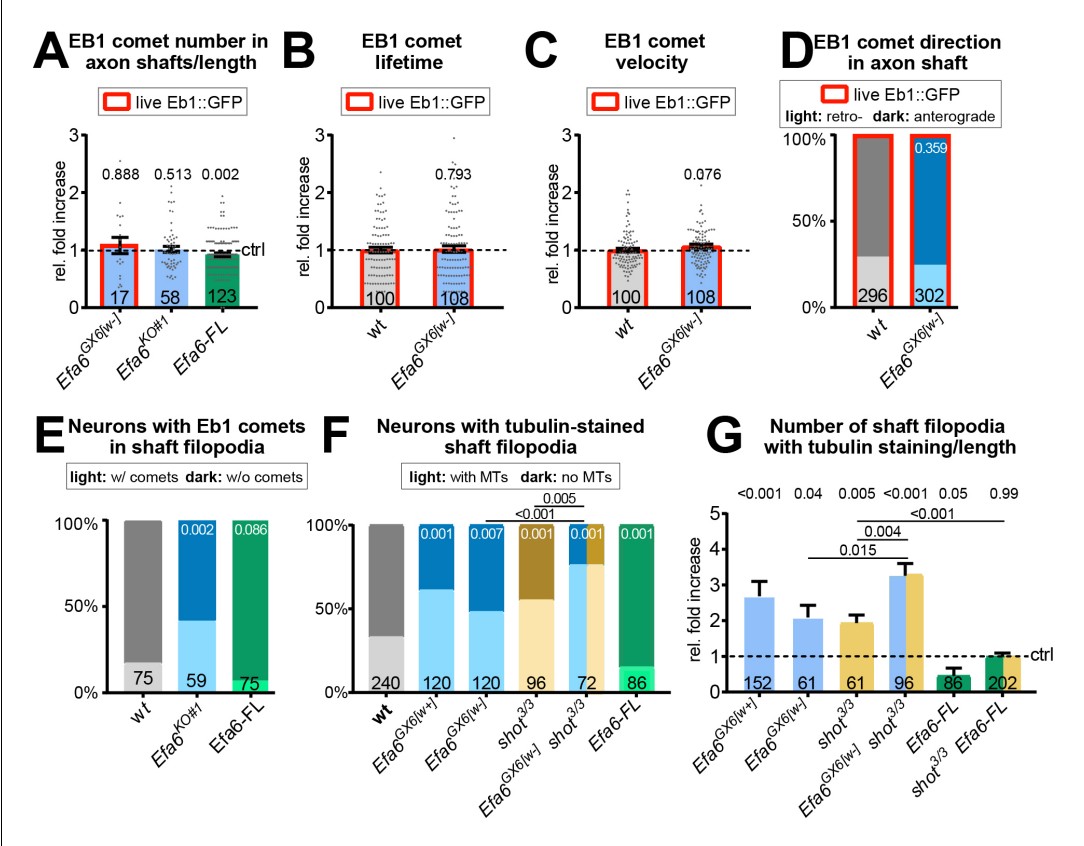

**Figure 6.** Loss of Efa6 promotes MT entry into axon shaft filopodia. Quantitative analyses of MT behaviours in axon shafts, as indicated above each graph; bars are colour-coded: grey, controls; blue, different *Efa6* mutant alleles; green, neurons over-expressing Efa-FL::GFP or Efa6::CAAX::GFP; orange, *shot*[3] mutant allele; red outlines indicate live imaging data, all others were obtained from fixed specimens. (A–C,G) Fold-changes relative to wild-type controls (indicated as horizontal dashed 'ctrl' line) shown as single data points and a bar indicating mean ± SEM; P values were obtained via Mann-Whitney tests; control values (dashed line) in B and C equate to an Eb1 comet lifetime of 7.18 s ± 0.35 SEM and a velocity of 0.169 µm/s ± 0.01 SEM; in G the number of shaft filopodia per neuron was divided by the axon length of that same neuron. (D–F) Binary parameters (light *versus* dark shades as indicated) provided as percentages; P values are given relative to control or between different genotypes (as indicated by black lines); they were obtained via Chi[2] tests. Numbers at the bottom of bars indicate sample numbers pooled from at least two replicates, respectively; data points reflect individual Eb1::GFP comets (A–D) or individual neurons (E–G). For raw data see *Figure 6—source data 1*.

The online version of this article includes the following source data for figure 6:

**Source data 1.** Summary of the statistics from *Figure 6*.

*tubulin84B-GFP* either alone (*GMR-tub* controls), or together with *Efa6[RNAi]* to knock down *Efa6* specifically in these neurons (*GMR-tub-Efa6[IR]*; see Materials and methods for details).

When analysing aged flies at 26–27 days, we found that *Efa6* knock-down caused a doubling in the occurrence of axonal swellings with disorganised axonal MTs: the average of total swellings per column section was increased from 0.3 in controls to 0.65 swellings upon Efa6 knock-down; about a third of these contained disorganised MTs (*GMR-tub-Efa6[IR]*: 0.23 per column section; *GMR-tub*: 0.13; *Figure 9*). These data demonstrated that our findings in cultured neurons are relevant in vivo.

We propose therefore that Efa6 provides a quality control mechanism that prevents MT disorganisation by inhibiting only MTs that have escaped axonal bundles. This model would also be consistent with the slow onset and gradual increase of MT disorganisation we observed upon Efa6 deficiency (*Figure 8I,J*).

## Efa6 and shot promote MT bundles through complementary mechanisms

If Efa6 provides a quality control mechanism that 'cleans up' explorative MTs, it should act in complementary ways with other factors that 'prevent' explorative MTs by actively keeping them in

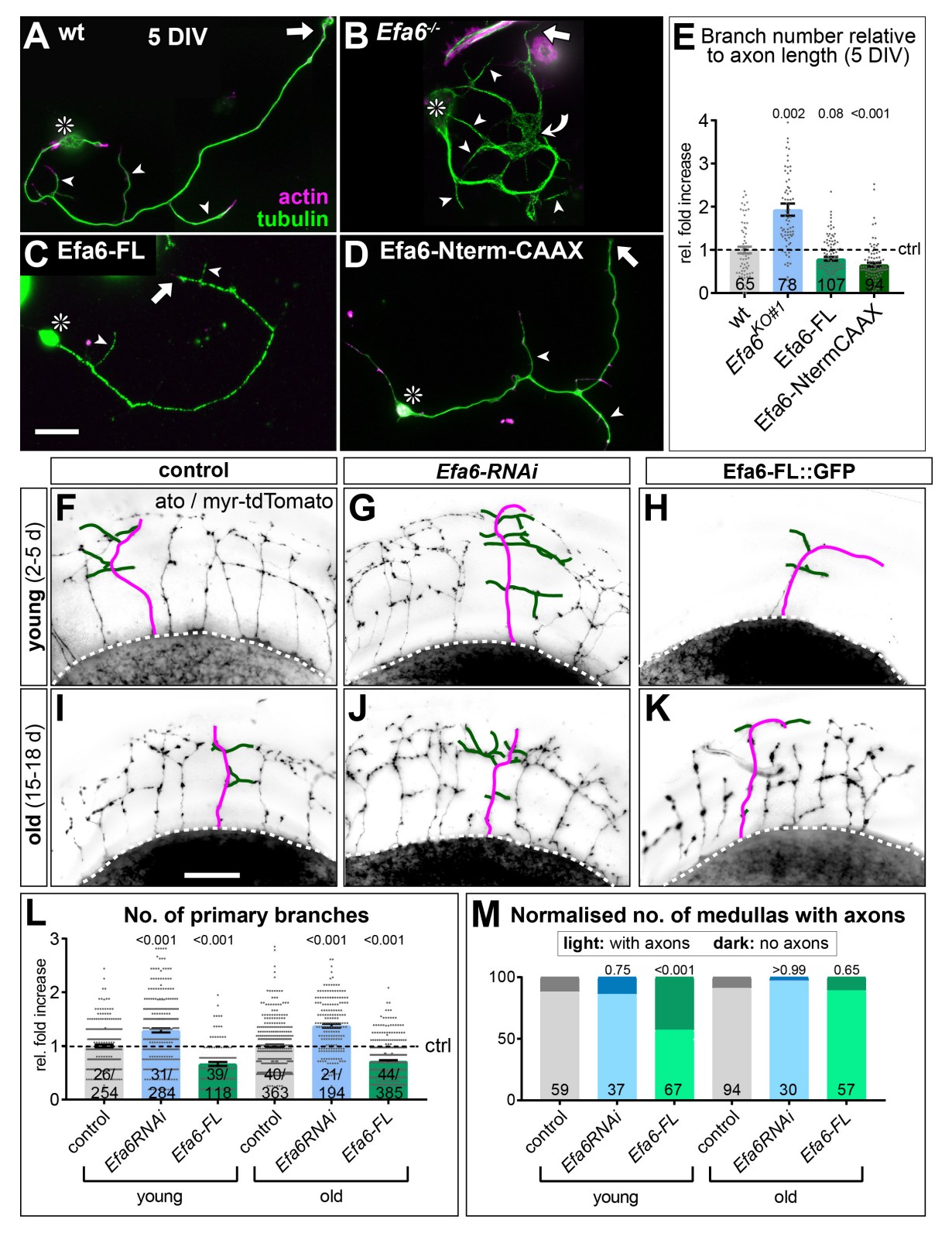

**Figure 7.** Efa6 regulates axon branching in primary *Drosophila* neurons and adult fly brains. (**A–D**) Examples of primary *Drosophila* neurons at 5DIV, all stained for actin (magenta) and tubulin (green); neurons are either wild-type controls (**A**), Efa6-deficient (**B**), expressing Efa6-FL::GFP (**C**), or expressing Efa6-Nterm::CAAX::GFP (**D**); asterisks indicate cell bodies, arrows point at axon tips, arrow heads at axon branches, the curved arrow at an area of MT disorganisation; the scale bar in C represents 20 μm in A-D. (**E**) Quantification of axonal branch numbers; different genotypes are colour-coded: grey,

*Figure 7 continued on next page*

*Figure 7 continued*

wild-type controls; blue, *Efa6* loss-of-function; green, neurons over-expressing Efa6 variants; data represent fold-change relative to wild-type controls (indicated as horizontal dashed 'ctrl' line); each neuron is shown as a single data point (sample number at bottom of bar) together with the mean ± SEM; P values from Mann-Whitney tests comparing to wild-type are given above each column. (F–K) Brains (medulla region of the optic lobe in oblique view) of young (2–5 d after eclosure; top) and old flies (15–18 d; bottom) driving *UAS-myr-tdTomato* via the *ato-Gal4* driver in dorsal cluster neurons (example neurons are traced in magenta for axons and green for side branches); flies either carry *ato-Gal4-*and *UAS-myr-tdTomato*, alone (control, left), together with *Efa6-RNAi* (middle) or together with Efa6-FL::GFP (right). (L,M) Quantification of data for control (wt; grey), Efa6 knock-down (blue) and Efa6-FL::GFP over-expression (green): L) shows the number of primary branches per axon as fold-change normalised to controls (indicated as horizontal dashed 'ctrl' line); individual axons are shown as data points (sample number at bottom of bars indicate number of medullas before and number of axons after slash); bars indicate mean ± SEM accompanied by single data points. (M) displays the number of medullas (sample numbers at bottom of bars) which display axons (light colours) or lack axons (dark colours) shown as a percentages in young and old flies. P values above columns were obtained from Mann-Whitney (L) or Chi$^2$ tests (M). Scale bar in I represents 60 μm in F-K. For raw data see *Figure 7—source data 1*.

The online version of this article includes the following source data and figure supplement(s) for figure 7:

**Source data 1.** Summary of the statistics from *Figure 7*.
**Figure supplement 1.** *ato-Gal4*-driven Efa-FL::GFP expression in adult brain tissue.

axonal bundles. Important preventive factors in both mammals and fly are the spectraplakins (*Bernier and Kothary, 1998*; *Dalpé et al., 1998*; *Voelzmann et al., 2017*). In *Drosophila*, spectra-plakins are represented by the single *short stop* (*shot*) gene; *shot* deficiency causes a severe increase in axonal off-track MTs and MT disorganisation (*Alves-Silva et al., 2012*; *Qu et al., 2017*; *Sanchez-Soriano et al., 2009*).

To study potential mutual enhancement of *Efa6* and *shot* mutant phenotypes, we first determined numbers of MTs in axonal shaft filopodia: both single-mutant conditions showed a strong enhance-ment of filopodial MTs (blue *vs.* orange bars in *Figure 6F,G*); this phenotype was substantially fur-ther increased in *shot³ Efa6^{GX6[w-]}* double-mutant neurons (orange/blue bars in *Figure 6F,G*). Vice versa, when transfecting *Efa6-FL-GFP* to boost the hypothesised 'cleaning-up' function, the *shot³* mutant phenotype was significantly improved (*Figure 6G*).

We then tested whether this increase in off-track MTs would correlate with more MT disorganisa-tion. At 6 HIV, *shot³* mutant neurons displayed a 2.4-fold, and *Efa6^{GX6[w-]}* mutant neurons a 1.55-fold increase in MDI (normalised to wild-type); this value was dramatically increased to 6.16 fold in *shot³ Efa6^{GX6[w-]}* double mutant neurons (*Figure 8M*). This suggests that Efa6 and Shot do not act through the same mechanism, but perform complementary roles in regulating and maintaining axonal MTs and MT bundles. This conclusion was further confirmed by our finding that transfection of *Efa6-FL-GFP* into *shot³* mutant neurons could alleviate the MDI phenotype (*Figure 8N*).

Finally, we assessed whether these complementary relationships between Shot and Efa6 are rele-vant in vivo. Since complete loss of Shot is an embryonically lethal condition, we first tested in cul-ture whether the lack of just one copy of *shot* has an enhancing effect on Efa6 deficiency. We found that MT disorganisation phenotypes of *Efa6-RNAi* (blue bar in *Figure 8L*) and of *shot^{3/+}* heterozy-gous mutant neurons (orange bar) at 6 HIV were clearly enhanced when both genetic manipulations were combined (orange/blue bar). When testing the same genetic constellations in our optic lobe model, we found that the originally observed increase in MT disorganisation caused by cell-autono-mous knock-down of *Efa6* (black arrows and blue bar in *Figure 9B,E*) was also further enhanced when the same experiment was carried out in a *shot^{3/+}* heterozygous mutant background (black arrows and orange/blue bar in *Figure 9C,E*).

These findings support our conclusion that there is a correlation between off-track MTs and MT disorganisation. Furthermore, they are consistent with a scenario where both Shot and Efa6 regulate axonal MTs but through independent and complementary pathways: Efa6 inhibits MTs at the cortex (with peripheral MTs persisting for longer if Efa6 is absent), whereas Shot actively maintains MTs in bundles (with more MTs going off-track if Shot is absent) - and both these functions complement each other during MT bundle maintenance and enhance each other's mutant phenotypes in double-mutant conditions (see further details in the Discussion).

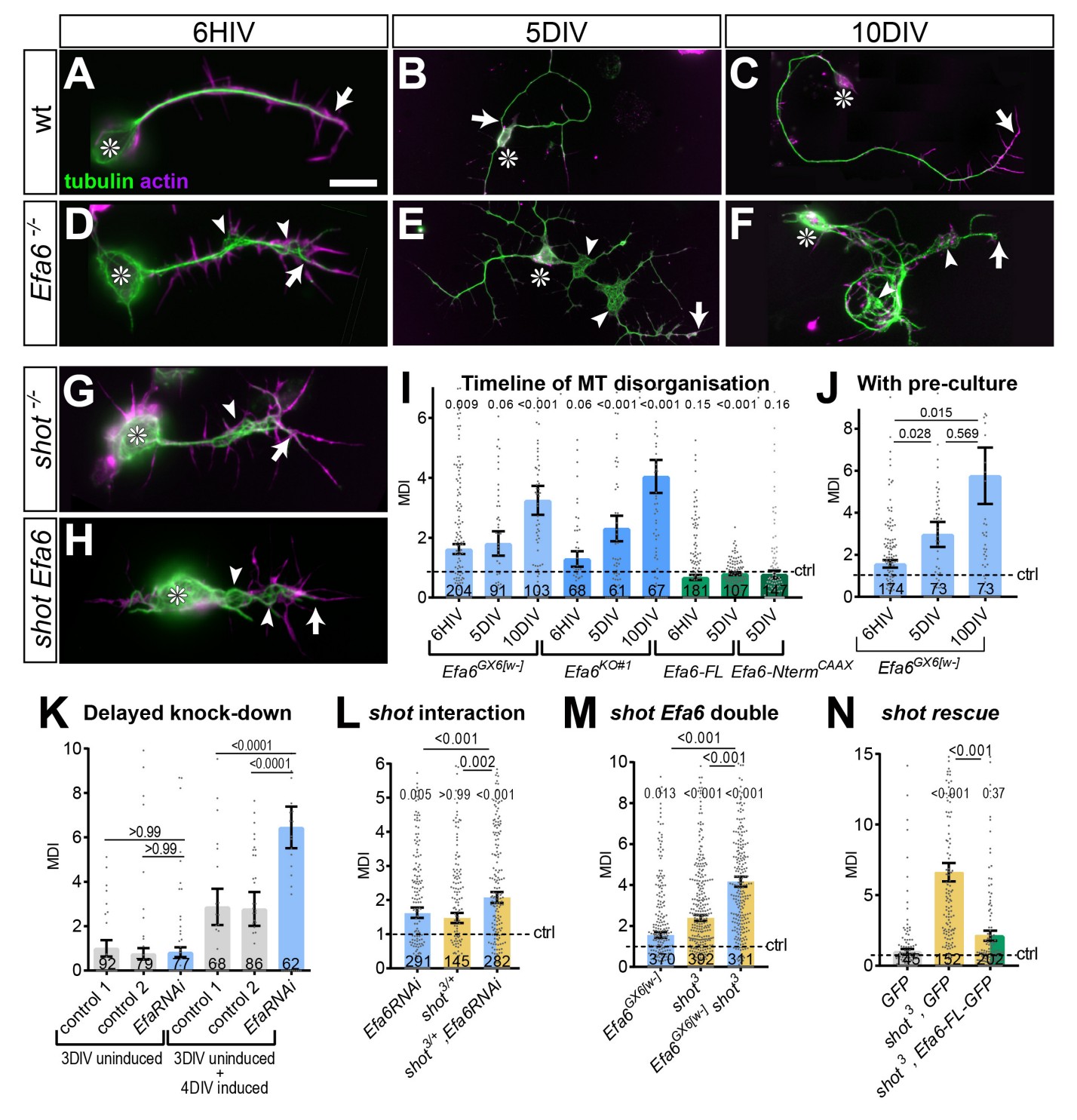

**Figure 8.** Efa6 helps to maintain axonal MT bundles in *Drosophila* neurons. (A–H) Images of primary neurons at 6HIV (left), 5DIV (middle) and 10DIV (right), stained for tubulin (green) and actin (magenta), derived from embryos that were either wild-type (wt, A–C), *Efa6* null mutant (D–F), homozygous for *shot³* (G) or *shot³ Efa6^{GX6[w-]}* double-mutant (shot Efa6, H; arrows point at axon tips, arrow heads at areas of MT disorganisation, and asterisks indicate the cell bodies; the scale bar in A represents 10 µm for 6HIV neurons and 25 µm for 5DIV and 10DIV neurons. (I–N) Quantitative analyses of MT disorganisation (measured as MT disorganisation index, MDI) in different experimental contexts (as indicated above graphs); different genotypes are colour-coded: grey, controls; blue, *Efa6* loss-of-function; orange, *shot³* in hetero-/homozygosis; green, neurons over-expressing Efa-FL::GFP or Efa6-Nterm::CAAX::GFP; in all cases, individual neurons are shown as single data points (sample numbers at bottom of bar), and the bars indicate mean ± SEM, all representing fold-change relative to wild-type controls (indicated as horizontal dashed 'ctrl' line); P values are given above each

*Figure 8 continued on next page*

*Figure 8 continued*

column and were obtained from Mann-Whitney tests (**I, J**) and Kruskal–Wallis one-way ANOVA with *post hoc* Dunn's test (**K–N**) either relative to controls or between genotypes (indicated by black lines). In K, 'control 1' is *tub-Gal80, elav-Gal4* alone and 'control 2' is *UAS-Efa6RNAi* alone. For raw data see *Figure 8—source data 1*.

The online version of this article includes the following source data for figure 8:

**Source data 1.** Summary of the statistics from *Figure 8*.

## Discussion

### Cortical collapse factors are important microtubule regulators relevant for axon morphology

Axons are the structures that wire our brain and body and are therefore fundamental to nervous system function. To understand how axons are formed during development, can be maintained in a plastic state thereafter, and why they deteriorate in pathological conditions, we need to improve our knowledge of axonal cell biology (*Salvadores et al., 2017*; *Sheng, 2017*). The MT bundles that form

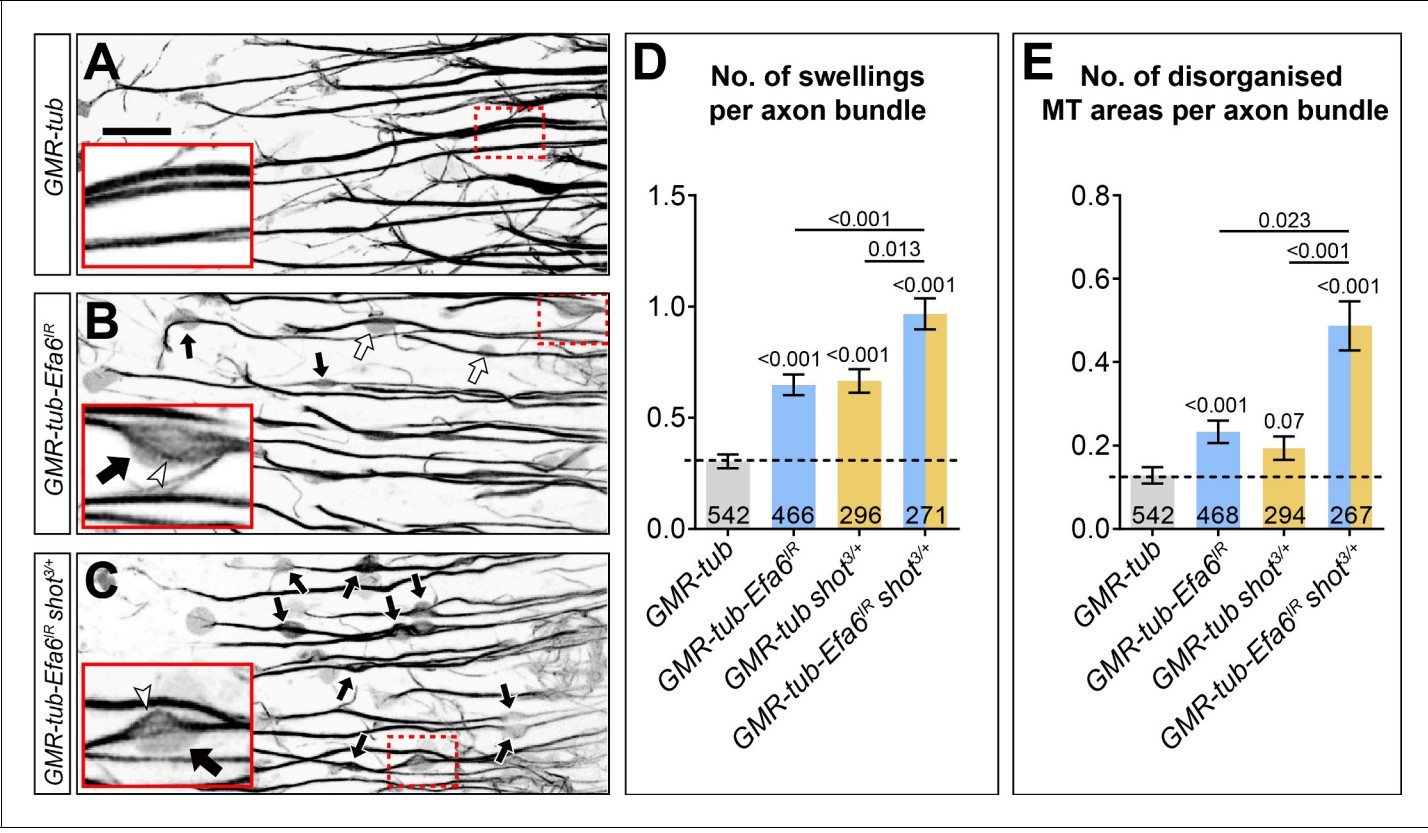

**Figure 9.** Efa6 is required for axonal MT bundle maintenance in adult fly brains. (**A–C**) Medulla region of adult brains at 26–27 days after eclosure, all carrying the *GMR31F10-Gal4* driver and *UAS-GFP-α-tubulin84B* (*GMR-tub*) which together stain MTs in a subset of lamina neuron axons that terminate in the medulla. The other specimens in addition co-express *Efa6-RNAi* either in wild-type background (*GMR-tub-Efa6^{IR}*) or in *shot^{3/+}* heterozygous mutant background (*GMR-tub-Efa6^{IR} shot^{3/+}*). White/black arrows indicate axonal swellings without/with MT disorganisation; rectangles outlined by red dashed lines are shown as 2.5 fold magnified insets where white arrow heads point at disorganised MTs; the scale bar in A represents 15 μm in all images. (**D, E**) Quantitative analyses of all axonal swelling (**D**) or swellings with MT disorganisation (**E**); different genotypes are colour-coded (grey, control; blue, Efa6 loss-of-function; orange, *shot^3* heterozygous); bars show mean ± SEM, all representing fold-change relative to wild-type controls (indicated as horizontal dashed line). P values from Kruskal–Wallis one-way tests are given above each column, sample numbers (i.e. individual axon bundles) at the bottom of each bar. For raw data see table *Figure 9—source data 1*.

The online version of this article includes the following source data for figure 9:

**Source data 1.** Summary of the statistics from *Figure 9*.

the core of axons are an essential aspect of this cell biology, and understanding how these bundles are regulated and contribute to axon morphogenesis will provide essential insights into axon development and maintenance (*Hahn et al., 2019*; *Voelzmann et al., 2016a*). Here, we have addressed fundamental contributions made by cortical collapse factors. We started from reports that two such factors from distinct protein families both negatively impact on axon growth in species as diverse as *C. elegans* (*Ce*Efa6; *Chen et al., 2015*; *Chen et al., 2011*) and mouse (Kif21A; *van der Vaart et al., 2013*).

We found that *Dm*Efa6 likewise acts as a negative regulator of axon growth. We demonstrate that fly Efa6 is a cortical collapse factor, inhibiting MTs primarily via the 18aa long MTED. Since the MTED is the only shared motif with *Ce*Efa6 in an otherwise entirely divergent N-terminus (*Figure 3C*), this clearly demonstrates that the MTED is functionally conserved between both species (*Chen et al., 2015*; *Chen et al., 2011*; *O'Rourke et al., 2010*).

Capitalising on *Drosophila* neurons as a conceptually well-established model for studies of axonal MT regulation (*Hahn et al., 2019*; *Prokop et al., 2013*), we went on to demonstrate two novel roles for Efa6: as a negative regulator of axon branching and a quality control factor maintaining MT bundle organisation. To perform these functions, Efa6 does not affect the dynamics of MTs contained within the central axonal bundles, but it inhibits mainly those MTs that leave these bundles (*Figure 10A*). By inhibiting explorative MTs in GCs, it negatively impacts on a key event underlying axon growth (explained below; yellow arrows in *Figure 10C*). By inhibiting off-track MTs in the axon shaft, it tones down the machinery that seeds new interstitial branches (red arrow in *Figure 10C*), but also prevents these MTs from going astray and causing MT disorganisation (curled MTs in *Figure 10C*).

Therefore, our work provides conceptual understanding of cortical collapse factors, which can explain how their molecular functions and subcellular roles in MT regulation link to their reported axonal growth phenotypes during development and regeneration (*Chen et al., 2015*; *Chen et al., 2011*; *Heidary et al., 2008*; *van der Vaart et al., 2013*), and to their additional functions in axon branching and maintenance reported here. Apart from existing links of cortical collapse factors to neurodevelopmental disorders (*Heidary et al., 2008*; *Tiab et al., 2004*; *van der Vaart et al., 2013*), we would therefore predict future links also to neurodegeneration.

## Roles of Efa6 during axonal growth

During axon growth, MTs constantly polymerise towards the periphery of GCs; the advance of many of these MTs is inhibited at the leading edge, and our work shows that cortical collapse factors are key mediators to this end. Only a fraction of MTs enters filopodia, potentially helped by active guidance mechanisms such as MT-actin cross-linkage (e.g. through spectraplakins, tau, drebrin-EB3; *Alves-Silva et al., 2012*; *Biswas and Kalil, 2018*; *Geraldo et al., 2008*). The widely accepted protrusion-engorgement-consolidation model of axon growth proposes that stabilised MTs in filopodia can seed axon elongation events (*Aletta and Greene, 1988*; *Goldberg and Burmeister, 1986*; *Prokop et al., 2013*). This model is consistent with our findings for Efa6. Thus loss of Efa6 can contribute to enhanced axon growth in two ways: firstly, through allowing more MTs to enter filopodia; secondly, by allowing them to dwell in filopodia for longer, thus enhancing the likelihood of their stabilisation (yellow arrows in *Figure 10C*). This scenario can explain why loss of *Efa6* in *C. elegans* improves axon re-growth after injury and growth overshoot during development (*Chen et al., 2015*; *Chen et al., 2011*), and why the higher levels of Kif21A levels in GCs causes stalled axon growth (*van der Vaart et al., 2013*).

In *C. elegans* it was shown that axonal injury leads to a re-localisation of *Ce*Efa6 to MT minus ends in the axon core (*Chen et al., 2015*). None of the conditions used in our study reproduced such behaviour with fly Efa6. Furthermore, it was shown that such central pools of *Ce*Efa6 require their MTED to recruit two kinases: TAC-1 (homologue of TACC/transforming-acidic-coiled-coil) and ZYG-8 (homologue of DCLK/Doublecortin-Like-Kinase; *Chen et al., 2015*). However, in contrast to Efa6, both of these kinases perform growth-enhancing functions and play a secondary, delayed role downstream of Efa6. They are therefore unsuited to explain the direct MT-inhibiting roles of the MTED (*Figure 4*). In contrast, virtually all structure-function analyses performed with *Ce*Efa6 in developing and regenerating axons perfectly match our data and can be explained through our proposed model. Based on our findings, one might argue that *Ce*Efa6 detachment from the membrane (*Chen et al., 2015*) could be the consequence of injury-induced physiological changes that would

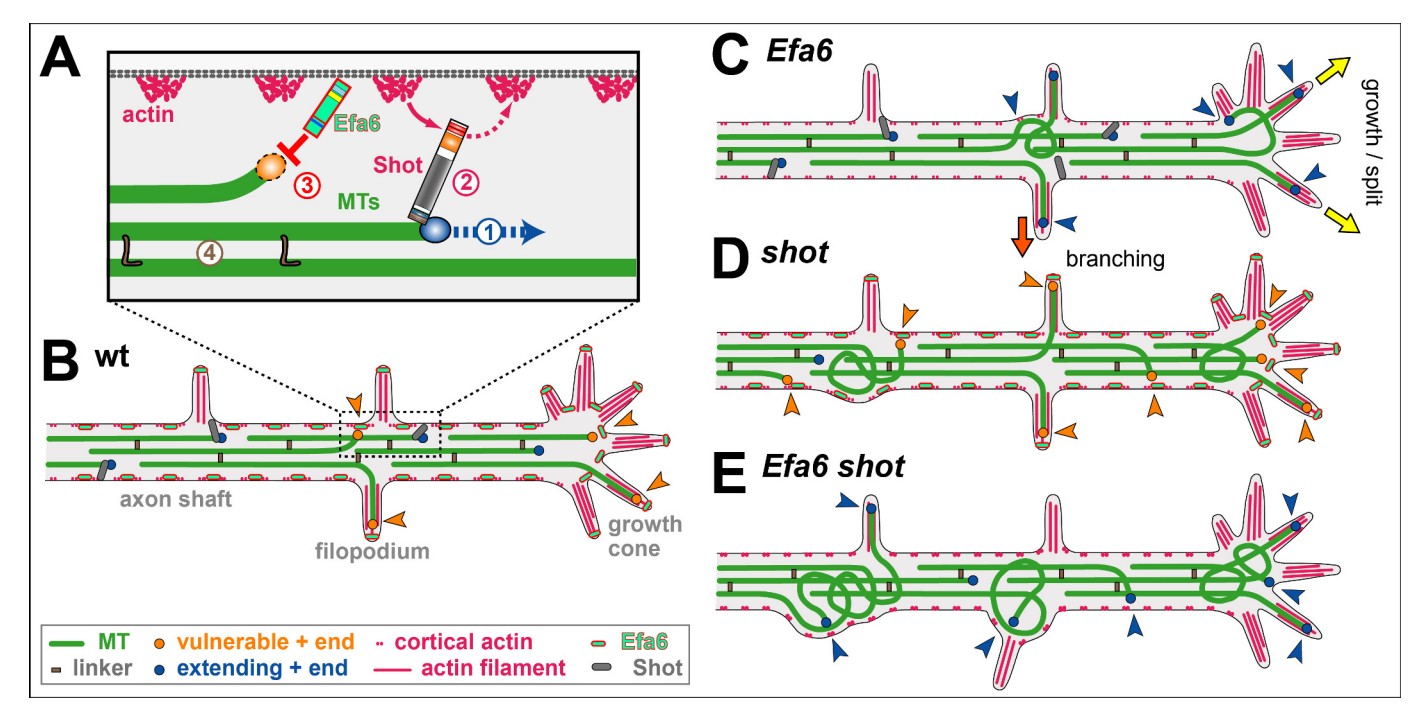

**Figure 10.** A model for axonal roles of Efa6. (**A**) The model of local axon homeostasis (*Hahn et al., 2019*; *Prokop, 2016*) states that the maintenance of axonal MT bundles (green bars) is an active process. For example, the polymerisation (1) mediated by plus end machinery (blue circle) is guided by spectraplakins (here Shot) along cortical actin into parallel bundles (2), or MTs are kept together through cross-linkage (brown 'L'; 4; *Bettencourt da Cruz et al., 2005*; *Krieg et al., 2017*); here we propose that MTs accidentally leaving the bundle become susceptible to inhibition through cortically anchored Efa6 (red 'T' and orange circle). (**B**) In normal neurons, MTs that polymerise within the axonal bundles (dark blue circles) are protected by Shot (grey lines) and MT-MT cross-linkers (brown rectangles), whereas MTs approaching the membrane (orange arrow heads) either by splaying out in GCs or leaving the bundle in the shaft (orange arrow heads) become susceptible (orange circles) to inhibition by Efa6 (light green/red dashes) both along the cortex and in filopodia. C) Upon Efa6 deficiency, MTs leaving bundles or entering GCs are no longer subjected to Efa6-mediated cortical inhibition (blue arrow heads) and can cause gradual build-up of MT disorganisation; when entering shaft filopodia they can promote interstitial branch formation (red arrow), when entering GC filopodia they can promote axon growth or even branching through GC splitting (yellow arrows). (**D**) Far more MTs leave the bundles in *shot* mutant neurons, but a good fraction of them can potentially be inhibited by Efa6 (increased number of orange arrow heads). (**E**) In the absence of both Shot and Efa6, more MTs leave the bundles, but there is no compensating cortical inhibition (increased number of blue arrow heads), so that the MT disorganisation phenotype worsens.

then pose a threat to axonal MT bundles; localisation to MT minus ends could therefore represent a protective sequestration mechanism. Another *C. elegans* study reported that loss of Efa6 has no impact on MT length in developing axons (*Yogev et al., 2016*), which appears consistent with our data (*Figure 6A–D*). They also found an increase in MT numbers, but there is currently no mechanism to explain this in non-injury conditions where *Ce*Efa6 stays at the membrane (*Chen et al., 2015*).

Interestingly, mammalian Efa6 also plays a role in axon regeneration. However, this mechanism is entirely different, in that it requires the C-terminus to activate Arf6 which, in turn, regulates integrin trafficking at the axon initial segment (*Eva et al., 2017*).

## Roles of Efa6 during axonal branching

Axon branching can occur via GC split, in that diverging MTs get stabilised in parallel in the same GC (*Acebes and Ferrús, 2000*; yellow arrows in *Figure 10C*). Alternatively, it can occur through interstitial branching which involves the active generation (e.g. through MT severing) and then stabilisation of off-track MTs (*Kalil and Dent, 2014*; *Lewis et al., 2013*; *Tymanskyj et al., 2017*; *Yu et al., 2008*). Both models agree with our observations in Efa6-deficient/over-expressing neurons: we find greater/lower numbers of MTs in GC and shaft filopodia at 6 HIV, which then correlate with enhanced/reduced axonal branch numbers in mature neurons (red arrow in *Figure 10C*).

If interstitial branch formation is negatively regulated by *Efa6*, this poses the question as to whether Efa6 has to be actively down-regulated in healthy neurons for branching to occur. Efa6 could either be physically removed from future branch points (*Chen et al., 2015*) or its MT inhibition function could be switched off. However, in our view, no such regulation is required because Efa6 seems to be in a well-balanced equilibrium. Enough Efa6 appears to be present to inhibit occasional, likely accidental off-track MTs; this capacity is surpassed when the number of off-track MTs is actively increased, for example through MT severing proteins during axonal branch formation (*Yu et al., 2008*). Such a saturation model is supported by our experiments with *shot* (*Figure 6F,G*): filopodial MT numbers are elevated in *shot* mutant neurons, although Efa6 is present and functional (as demonstrated by the further increase in filopodial MT numbers in *shot Efa6* double-mutant neurons; *Figure 10D,E*). This is consistent with a model where Efa6 function occurs at a level that is easily saturated when increasing the number of explorative MTs. Such a view would also explain why loss of *Ce*Efa6 promotes axon regeneration in *C. elegans* (*Chen et al., 2015*; *Chen et al., 2011*), in that the constant base-line of MT inhibition present in the wild-type, is removed in the mutant condition, thus favouring growth-mediating explorative MTs.

## Roles of Efa6 during axonal MT bundle maintenance

Axonal MT disorganisation in Efa6-deficient neurons occurs gradually and can even be induced by knock-down of Efa6 at mature stages (*Figure 8K*). Therefore, Efa6 appears to prevent MT disorganisation during axon development and maintenance, as is consistent with its continued expression in the nervous system (*Figure 1*). Such a continued role makes sense in a scenario where MT bundles remain highly dynamic throughout a neuron's lifetime, constantly undergoing polymerisation to drive renewal processes that prevent senescence (*Hahn et al., 2019*; *Voelzmann et al., 2016a*).

Based on these findings, we propose that Efa6 acts as a quality control or maintenance factor within our model of 'local axon homeostasis' (*Hahn et al., 2019*; *Prokop, 2016*). This model states that MTs in the force-enriched environment of axons have a tendency to go off-track and curl up (*Pearce et al., 2018*), thus potentially seeding MT disorganisation. Different classes of MT-binding regulators, amongst them spectraplakins, prevent this by actively promoting the bundled conformation (*Hahn et al., 2019*). We propose that cortical collapse factors act in a complementary way to spectraplakins in that they play no role in maintaining MTs in bundles, but they inhibit those MTs that have escaped the bundling mechanisms (*Hahn et al., 2019*).

In this scenario, MTs are protected from cortical collapse as long as they are actively maintained in axonal bundles; this can explain the long known conundrum of how axonal MTs extend hundreds of micrometres in relative proximity to the cell cortex in axons, whereas in non-neuronal cells cortical proximity of MTs tends to trigger either their inhibition or tethered stabilisation (*Fukata et al., 2002*; *Kaverina et al., 1998*).

## Evolutionary and mechanistic considerations of Efa6 function

We found that the MTED motif correlates well with MT inhibiting functions of Efa6 family members, whereas the rest of the N-terminus bears no obvious further similarity. Our experiments with N-terminal protein and synthetic MTED peptide, both reveal association with MTs/tubulin. The MTED strongly interferes with MT polymerisation. Future co-crystallisation experiments are required to reveal how the MTED works. Given its small size we hypothesise that it simply blocks assembly, rather than acting via more complex mechanisms such as active promotion of depolymerisation (e.g. kinesin-8 and −13, XMap215; *Al-Bassam and Chang, 2011*; *Brouhard and Rice, 2014*) or severing (e.g. spastin, katanin, fidgetin; *McNally and Roll-Mecak, 2018*; *Sharp and Ross, 2012*).

In any case, the small size of MTEDs might come in handy as experimental tools to inhibit MTs, potentially displaying complementary properties to existing genetic tools such as the kinesin-13 Kif2C (*Moore et al., 2005*; *Schimizzi et al., 2010*), stathmin (*Marklund et al., 1996*) or spastin (*Eckert et al., 2012*). Importantly, the experiments with the CAAX domain have shown that Efa6's MT inhibiting function can be targeted to specific subcellular compartments to clear them of MTs, thus opening up a wide range of future applications.

Interestingly, the MT-inhibiting role of Efa6 seems not to be conserved in chordates when taking the MTED as indicator for this function (*Figure 3—figure supplement 4A*). However, roles of cortical collapse factors in neurons seem to have been taken over by other proteins such as the kinesin-4

family member Kif21A. The CFEOM1-linked Kif21A[R954W] mutation causes the protein to relocate from the axon shaft to the growth cone of cultured hippocampal neurons (*van der Vaart et al., 2013*). In consequence, increased Kif21A levels in GCs cause reduced axon growth - and we observed the same with Efa6 over-expression (green bars in *Figure 2D*). The decreased levels of Kif21A in proximal axons correlate with a local increase in side branches - and we observed the same with Efa6 loss of function (blue bars in *Figure 7E,L*).

Finally, we found that the C-terminal domains of Efa6 might display some degree of functional conservation. So far, work on mammalian PSDs has revealed functions for C-terminal domains in regulating ARF6, ARF1 or ARL14 during actin cytoskeletal reorganisation and membrane ruffling, tumour formation, axon regeneration and immune regulation (*Derrien et al., 2002*; *Eva et al., 2017*; *Paul et al., 2011*; *Pils et al., 2005*). Our finding that PSD1 and C-terminal Efa6 constructs cause similar membrane ruffling phenotypes in fibroblasts (*Figure 3—figure supplements 6* and *7*), suggests that some conserved functions reside in this region and might further contribute, together with N-terminally mediated MT inhibition, to the neuronal or non-neuronal defects that cause semi-lethality displayed by *Efa6* mutant flies (data not shown).

## Conclusions and future perspectives

We propose that Efa6 acts as a cortical collapse factor which is important for the regulation of axonal MTs and relevant for axon growth, maintenance and branching. Although this function of Efa6 is evolutionarily not widely conserved, our findings provide a helpful paradigm for studies of other classes of cortical collapse factors also in mammalian neurons. Promising research avenues will be to refine our mechanistic understanding of how Efa6 blocks MT polymerisation, not only to better understand how it can be regulated in axons, but also to better exploit MTEDs as molecular tools in cell biological research.

## Materials and methods

### Fly stocks

Loss-of-function mutant stocks used in this study were the two deficiencies uncovering the *Efa6* locus *Df(3R)Exel6273* (94B2-94B11 or 3:22,530,780..22,530,780; RRID:BDSC_7740) and *Df(3R)ED6091* (94B5-94C4 or 3R:22,587,681..22,587,681; RRID:DGGR_150165): *shot[3]* (the strongest available allele of *short stop*; *Kolodziej et al., 1995*; *Sanchez-Soriano et al., 2009*; RRID:BDSC_5141); *Efa6[KO#1]* (ends-out targeting mutant that contains a small 74 bp deletion in exon 8) and three null alleles generated as genomic engineering intermediates: the knock-out founder line *Efa6[GX6[w+]]* and the two founder lines *Efa6[GX6[w-]]* (RRID:BDSC_60587) and *Arf51F[GX16[w-]]* (RRID:BDSC_60585; all published in *Huang et al., 2009*). Gal4 driver lines used were the pan-neuronal lines *sca-Gal4* (strongest in embryos; *Sánchez-Soriano et al., 2010*) and *elav-Gal4* (1[st] and 3[rd] chromosomal, both expressing at all stages; RRID:DGGR_105921, RRID:BDSC_8760; *Luo et al., 1994*), *GMR31F10-Gal4* (Bloomington #49685; RRID:BDSC_49685), as well as the *ato-Gal4* line expressing in a subset of neurons in the adult brain (*Hassan et al., 2000*; *Voelzmann et al., 2016b*); RRID:DGGR_108799). Lines for targeted gene expression were *UAS-Efa6[RNAi]* (VDRC #42321; RRID:FlyBase_FBst0464531), *UAS-Gal80[ts]* (*Zeidler et al., 2004*), *UAS-Eb1-GFP* (*Alves-Silva et al., 2012*), *UAS-GFP-α-tubulin84B* (*Grieder et al., 2000*) and *UAS-myr-tdTomato* (*Zschätzsch et al., 2014*; RRID:BDSC_32222). Efa6 expression was detected via the genomically engineered *Efa6-GFP* allele, where a GFP was inserted after the last amino acid in exon 14 (*Huang et al., 2009*).

### *Drosophila* primary cell culture

*Drosophila* primary neuron cultures were performed as published previously (*Prokop et al., 2012*). In brief, stage 11 embryos were treated for 1 min with bleach to remove the chorion, sterilized for ~30 s in 70% ethanol, washed in sterile Schneider's medium (Gibco) with 20% fetal calf serum (Gibco), and eventually homogenized with micro-pestles in 1.5 centrifuge tubes containing 21 embryos per 100 μl sterile filtered dispersion medium [167 ml distilled water, 30 ml Hanks' Balanced Salt Solution without calcium or magnesium (Gibco), 3 ml penicillin-streptomycin-solution (10,000 units; Gibco), 0.01 g phenyl-thio urea (Sigma), 0.5 mg/ml collagenase type 1 (Worthington, Cellsystems) and 2 mg/ml Dispase (Roche)] and left to incubate for 5 min at 37°C. Cells are washed with

Schneider's/FCS , spun down for 4 mins at 650 g, supernatant was removed and cells re-suspended in 90 µl of Schneider's/FCS. 30 µl drops were placed on cover slips. Cells were allowed to adhere for 90–120 min either directly on glass or on cover slips coated with a 5 µg/ml solution of concanavalin A, and then grown as a hanging drop culture for hours or days at 26°C as indicated.

To abolish maternal rescue of mutants, that is masking of the mutant phenotype caused by deposition of normal gene product from the healthy gene copy of the heterozygous mothers in the oocyte (*Roote and Prokop, 2017*), we used a pre-culture strategy (*Prokop et al., 2012*; *Sánchez-Soriano et al., 2010*) where cells were kept for 5 days in a tube before they were re-dispersed and plated on a coverslip.

For the transfection of *Drosophila* primary neurons, a quantity of 70–75 embryos per 100 µl dispersion medium was used. After the washing step and centrifugation, cells were re-suspended in 100 µl transfection medium [final media containing 0.1–0.5 µg DNA and 2 µl Lipofecatmine 2000 (L2000)]. To generate this media, dilutions of 0.1–0.5 µg DNA in 50 µl Schneider's medium and 2 µl L2000 in 50 µl Schneider's medium were prepared, then mixed together and incubated at room temperature for 5–30 mins, before being added to the cells in centrifuge tubes where they were kept for 24 hr at 26°C. Cells were then treated again with dispersion medium, re-suspended in culture medium and plated out as described above.

For temporally controlled knock-down experiments we used flies carrying the driver construct *elav-Gal4*, the knock-down construct *UAS-Efa6-RNAi*, and the temperature-sensitive Gal4 inhibitor *UAS-Gal80ts*, all in parallel. At the restrictive temperature of 19°C, Gal80ts blocks Gal4-induced expression of *Efa6-RNAi*, and this repressive action is removed at the permissive temperature of 27°C where Gal80ts is non-functional. Control neurons were from flies carrying only the *Gal4/Gal80* (control 1 in *Figure 8K*) or only the *Efa6-RNAi* transgene (control 2).

## Fibroblast cell culture

NIH/3T3 mouse fibroblast cells (ATCC; RRID:CVCL_0594) were grown in DMEM supplemented with 1% glutamine (Invitrogen), 1% penicillin/streptomycin (Invitrogen) and 10% FCS in culture dishes (100 mm with vents; Fisher Scientific UK Ltd) at 37°C in a humidified incubator at 5% $CO_2$. Cell lines have been tested for mycoplasma and are free of contamination. Cells were split every 2–3 d, washed with pre-warmed PBS, incubated with 4 ml of Trypsin-EDTA (T-E) at 37°C for 5 min, then suspended in 7 ml of fresh culture medium and eventually diluted (1/3-1/20 dilution) in a culture dish containing 10 ml culture media.

For transfection of NIH/3T3 cells, 2 ml cell solution (~$10^5$ cells per ml) was first transferred to 6-well plates, and grown overnight to double cell density. 2 µg of DNA and 2 µl Plus reagent (Invitrogen) were added to 1 ml serum-free media in a centrifuge tube, incubated for five mins at RT, then 6 µl Lipofectamine (Invitrogen) were added, and incubated at RT for 25 min. Cells in the 6-well plate were washed with serum-free medium and 25 min later DNA/Lipofectamine was mixed into the medium (1/1 dilution). Plates were incubated for 3 hr at 37°C, washed with 2 ml PBS, 400 µl trypsin were added for five mins (37°C), then 3 ml complete medium; cells were suspended and added in 1 ml aliquots to 35 mm glass-bottom dishes (MatTek) coated with fibronectin [300 µl of 5 µg/ml fibronectin (Sigma-Aldrich) placed in the center of a MatTek dish for 1 hr at 37°C, then washed with PBS]; 1 ml of medium was added and cells grown for 6 hr or 24 hr at 37°C in a $CO_2$ incubator. For live imaging, the medium was replaced with 2 ml Ham's F-12 medium + 4% FCS.

## Dissection of adult brains

To analyse the function of Efa6 in MT bundle integrity in medulla axons in vivo, flies were aged at 29°C. Flies were maintained in groups of up to 20 flies of the same gender (*Stefana et al., 2017*) and changed into new tubes every 3–4 days. Brain dissections were performed in Dulbecco's PBS (Sigma, RNBF2227) after briefly sedating them on ice. Dissected brains with their laminas and eyes attached were placed into a drop of Dulbecco's PBS on MatTek glass bottom dishes (P35G1.5–14C), covered by coverslips and immediately imaged with a 3i Marianas Spinning Disk Confocal Microscope.

To measure branching in *ato-Gal4 Drosophila* neurons, adult brains were dissected in Dulbecco's PBS and fixed with 4% PFA for 15 min. Antibody staining and washes were performed with PBT (PBS supplemented with 0.3% Triton X-100). Specimens were embedded in Vectashield (VectorLabs).

**Table 1.** Constructs and inserts used to generate expression and transgenic constructs.

The left column lists the final vector representing intermediate steps of cloning procedures or vectors employed for experiments on this study, the middle column the source of the vectors, and the right column the inserts of vectors on the left. Abbreviations: co = codon optimised; Dm = Drosphila melanogaster; Ce = Caenorhabditis elegans; Hs = Homo sapiens.

| final vector | Source | insert |
|---|---|---|
| pcDNA3-EGFP | Addgene | XhoI-EGFP-XbaI |
| pUAST-AscI-PacI-EGFP | this study | KpnI, AscI, PacI-EGFP-XbaI |
| pUAST-DmEfa6FL-EGFP (aa1-1387) | this study | KpnI, AscI-**kozak**-Efa6 (aa1-1387)-**GSGSGS-EGFP**-PacI, XbaI |
| P[acman]M-6-attB-UAS-1-3-4-DmEfa6FL-EGFP (aa1-1387) | this study | AscI-**kozak**-DmEfa6 (aa1-1387)-**GSGSGS-EGFP**-PacI |
| pcDNA3.1-DmEfa6FL-EGFP (aa1-1387) | this study | KpnI, AscI-**kozak**-DmEfa6 (aa1-1387)-**GSGSGS-EGFP**-PacI, XbaI |
| pUAST-DmEfa6ΔCterm-EGFP (aa1-894) | this study | KpnI, AscI-**kozak**-DmEfa6**ΔCterm** (aa1-894)-**GSGSGS-EGFP**-PacI, XbaI |
| P[acman]M-6-attB-UAS-1-3-4-DmEfa6ΔCterm-EGFP (aa1-894) | this study | AscI-**kozak**-DmEfa6**ΔCterm** (aa1-894)-**GSGSGS-EGFP**-PacI |
| pcDNA3.1-DmEfa6ΔCterm-EGFP (aa1-894) | this study | KpnI, AscI-**kozak**-DmEfa6**ΔCterm** (aa1-894)-**GSGSGS-EGFP**-PacI, XbaI |
| pUAST-DmEfa6-Nterm-EGFP (aa1-410) | this study | KpnI, AscI-**kozak**-DmEfa6-Nterm (aa1-410)-**GSGSGS-EGFP**-PacI, XbaI |
| P[acman]M-6-attB-UAS-1-3-4-DmEfa6-Nterm-EGFP (aa1-410) | this study | AscI-**kozak**-DmEfa6-Nterm (aa1-410)-**GSGSGS-EGFP**-PacI |
| pcDNA3.1-DmEfa6-Nterm-EGFP (aa1-410) | this study | KpnI, AscI-**kozak**-DmEfa6-Nterm (aa1-410)-**GSGSGS-EGFP**-PacI, XbaI |
| pUAST-DmEfa6-Nterm-CAAX-EGFP (aa1-410) | this study | KpnI, AscI-**kozak**-DmEfa6-Nterm (aa1-410)-**GSGSGS-EGFP-CAAX[KRAS]**-PacI, XbaI |
| P[acman]M-6-attB-UAS-1-3-4-DmEfa6-Nterm-CAAX-EGFP (aa1-410) | this study | AscI-**kozak**-DmEfa6-Nterm (aa1-410)-**GSGSGS-EGFP-CAAX[KRAS]**-PacI |
| pcDNA3.1-DmEfa6-Nterm-CAAX-EGFP (aa1-410) | this study | KpnI, AscI-**kozak**-DmEfa6-Nterm (aa1-410)-**GSGSGS-EGFP-CAAX[KRAS]**-PacI, XbaI |
| pUAST-DmEfa6-NtermΔSxiP-EGFP (aa1-410) | this study | KpnI, AscI-**kozak**-DmEfa6-Nterm**ΔSxiP** (aa1-410; **SQIP > AAAA; SRIP > AAAA**)-**GSGSGS-EGFP**-PacI, XbaI |
| pcDNA3.1-DmEfa6-NtermΔSxiP-EGFP (aa1-410) | this study | KpnI, AscI-**kozak**-DmEfa6-Nterm**ΔSxiP** (aa1-410; **SQIP > AAAA; SRIP > AAAA**)-**GSGSGS-EGFP**-PacI, XbaI |
| pUAST-DmEfa6-NtermΔMTED-EGFP (aa1-300) | this study | KpnI, AscI-**kozak**-DmEfa6-Nterm**ΔMTED** (aa1-300)-**GSGSGS-EGFP**-PacI, XbaI |
| pcDNA3.1-DmEfa6-NtermΔMTED-EGFP (aa1-300) | this study | KpnI, AscI-**kozak**-DmEfa6-Nterm**ΔMTED** (aa1-300)-**GSGSGS-EGFP**-PacI, XbaI |
| pUAST-DmEfa6ΔNerm-EGFP (aa851-1387) | this study | KpnI, AscI-**kozak**-DmEfa6**ΔNerm** (aa851-1387)-**GSGSGS-EGFP**-PacI, XbaI |
| pcDNA3.1-DmEfa6ΔNerm-EGFP (aa851-1387) | this study | KpnI, AscI-**kozak**-DmEfa6**ΔNerm** (aa851-1387)-**GSGSGS-EGFP**-PacI, XbaI |
| pUAST-DmEfa6-MTED-EGFP (aa322-341) | this study | KpnI, AscI-**kozak**-DmEfa6-MTED (aa322-341)-**GSGSGS-EGFP**-PacI, XbaI |
| pcDNA3.1-DmEfa6-MTED-EGFP (aa322-341) | this study | KpnI, AscI-**kozak**-DmEfa6-MTED (aa322-341)-**GSGSGS-EGFP**-PacI, XbaI |
| pcDNA3.1-CeEfa6-FL-EGFP (aa1-816) | this study | KpnI, AscI-**kozak**-CeEfa6 (aa1-816)-**GSGSGS-EGFP**-PacI, XbaI |
| pcDNA3.1-CeEfa6-Nterm-EGFP (aa1-152) | this study | KpnI, AscI-**kozak**-CeEfa6-Nterm (aa1-152)-**GSGSGS-EGFP**-PacI, XbaI |
| pcDNA3.1-CeEfa6-MTED-EGFP (aa24-42) | this study | KpnI, AscI-**kozak**-CeEfa6-MTED (aa24-42)-**GSGSGS-EGFP**-PacI, XbaI |

Table 1 continued

| final vector | Source | insert |
|---|---|---|
| *pcDNA3.1-HsPSD1-FL-EGFP (aa1-1024)* | this study | *KpnI, AscI*-**kozak**-*Hs*PSD1 (aa1-1024)-**GSGSGS**-*NotI*-**EGFP**-*PacI, XbaI* |
| *pcDNA3.1-HsPSD1-Nterm-EGFP (aa1-280)* | this study | *KpnI, AscI*-**kozak**-*Hs*PSD1-Nterm (aa1-280)-**GSGSGS**-*NotI*-**EGFP**-*PacI, XbaI* |
| *pcDNA3.1-HsPSD1-MTED-EGFP (aa31-49)* | this study | *KpnI, AscI*-**kozak**-*Hs*PSD1-MTED (aa31-49)-**GSGSGS**-*NotI*-**EGFP**-*PacI, XbaI* |
| *pcDNA3.1-HsPSD2-FL-EGFP (aa1-771)* | this study | *KpnI, AscI*-**kozak**-*Hs*PSD2 (aa1-771)-**GSGSGS**-*NotI*-**EGFP**-*PacI, XbaI* |
| *pcDNA3.1-HsPSD3-EGFP (aa515-1047)* | this study | *KpnI, AscI*-**kozak**-*Hs*PSD3 (aa515-1047)-**GSGSGS**-*NotI*-**EGFP**-*PacI, XbaI* |
| *pcDNA3.1-HsPSD4-FL-EGFP (aa1-1027)* | this study | *KpnI, AscI*-**kozak**-*Hs*PSD4 (aa1-1027)-**GSGSGS**-*NotI*-**EGFP**-*PacI, XbaI* |
| *pcDNA3.1-co-HsPSD1-MTED-EGFP (aa31-49)* | this study | *KpnI, AscI*-**kozak**-*Hs*PSD1-MTED (aa31-49)-**GSGSGS**-**EGFP**-*PacI, XbaI* |
| *pcDNA3.1-co-CeEfa6-MTED-EGFP (aa24-42)* | this study | *KpnI, AscI*-**kozak**-*Ce*Efa6-MTED (aa24-42)-**GSGSGS**-**EGFP**-*PacI, XbaI* |
| *pcDNA3.1-co-DmEfa6-MTED-EGFP (aa322-341)* | this study | *KpnI, AscI*-**kozak**-*Dm*Efa6-MTED (aa322-341)-**GSGSGS**-**EGFP**-*PacI, XbaI* |
| *pCS107-DmEfa6-Nterm-EGFP* | this study | *NotI*-**kozak**-*Dm*Efa6-Nterm (aa1-410)-**GSGSGS-EGFP**-*StuI* |
| *pFastBac-His6-MCAK-EGFP-StrepII* | not known | His6-MCAK::EGFP-StrepII |
| *pFastBac-His6-DmEfa6ΔCterm-EGFP-StrepII (aa1-894)* | this study | His6-*Dm*Efa6ΔCterm::EGFP-StrepII (aa1-894) |

## Immunohistochemistry

Primary fly neurons and fibroblasts were fixed in 4% paraformaldehyde (PFA) in 0.05 M phosphate buffer (PB; pH 7–7.2) for 30 min at room temperature (RT); for anti-Eb1 staining, ice-cold +TIP fix (90% methanol, 3% formaldehyde, 5 mM sodium carbonate, pH 9; stored at −80°C and added to the cells; *Rogers et al., 2002*) was added for 10 mins. Adult brains were dissected out of their head cases in PBS and fixed with 4% PFA in PBS for 1 hr, followed by a 1 hr wash in PBT.

Antibody staining and washes were performed with PBT. Staining reagents: anti-tubulin (RRID: AB_477593, clone DM1A, mouse, 1:1000, Sigma; alternatively, RRID:AB_2210391, clone YL1/2, rat, 1:500, Millipore Bioscience Research Reagents); anti-DmEb1 (gift from H. Ohkura; rabbit, 1:2000; *Elliott et al., 2005*); anti-Elav (mouse, 1:1000, DSHB, RRID:AB_528218); anti-GFP (ab6673, goat, 1:500, Abcam, RRID:AB_305643); Cy3-conjugated anti-HRP (goat, 1:100, Jackson ImmunoResearch); F-actin was stained with Phalloidin conjugated with TRITC/Alexa647, FITC or Atto647N (1:100 or 1:500; Invitrogen and Sigma). Specimens were embedded in ProLong Gold Antifade Mountant.

## Microscopy and data analysis

Standard documentation was performed with AxioCam monochrome digital cameras (Carl Zeiss Ltd.) mounted on BX50WI or BX51 Olympus compound fluorescent microscopes. For the analysis of *Drosophila* primary neurons, we used two well-established parameters (*Alves-Silva et al., 2012*; *Sánchez-Soriano et al., 2010*): axon length (from cell body to growth cone tip; measured using the segmented line tool of ImageJ) and the degree of MT disorganisation in axons which was either measured as binary score or ratio (percentage of neurons showing obvious MT disorganisation in their axons) or as 'MT disorganisation index' (MDI; *Qu et al., 2017*): the area of disorganisation was measured using the freehand selection in ImageJ; this value was then divided by axon length (see above) multiplied by 0.5 µm (typical axon diameter, thus approximating the expected area of the axon if it were not disorganised). For Eb1::GFP comet counts, neurons were subdivided into axon shaft and growth cones (GC): the proximal GC border was set where the axon widens up (broader GCs) or where filopodia density increases significantly (narrow GCs). MT loss in fibroblasts was assessed on randomly chosen images of successfully transfected, GFP-expressing fibroblasts, stained

for tubulin and actin. Due to major differences in plasma membrane versus cytoplasmic localisation of constructs, their expression strengths could not be standardised. Assuming a comparable expression strength distribution, we therefore analysed all transfected cells in the images and assigned them to three categories: 'MTs normal', 'MTs damaged' or 'prominent MT networks gone' (*Figure 3G–G''*). To avoid bias, image analyses were performed blindly, that is the genotype or treatment of specimens was masked. To analyse ruffle formation in fibroblasts, cells were stained with actin and classified (with or without ruffles).

To assess the degree of branching, we measured axonal projections of dorsal cluster neurons in the medulla, which is part of the optic lobe in the adult brain (*Hassan et al., 2000*; *Voelzmann et al., 2016b*). These neurons were labelled by expressing *UAS-myr-tdTomato* via the *ato-Gal4* driver either alone (control), together with *UAS-Efa6*[RNAi] or together with *UAS-Efa6-FL-GFP*. We analysed them in young brains (2–5 d after eclosure of flies from their pupal case) or old brains (15–18 d). Z-stacks of adult fly brains (optic lobe area) were taken with a Leica DM6000 B microscope and extracted with Leica MM AF Premier software. They were imaged from anterior and the number of branches was quantified manually. Branches were defined as the protrusions from the DC neuron axons in the medulla. Branches in fly primary neurons at 5DIV were also counted manually and defined as MT protrusions from main axon.

To measure MT disorganisation in the optic lobe of adult flies, *GMR31F10-Gal4* (Bloomington #49685) was used to express *UAS-GFP-α-tubulin84B* (*Grieder et al., 2000*) in a subset of lamina axons which project within well-ordered medulla columns (*Prokop and Meinertzhagen, 2006*). Flies were left to age for 26–27 days (about half their life expectancy) and then their brains were dissected out, mounted in MatTek dishes and imaged using a 3i spinning disk confocal system at the ITM Biomedical imaging facility at the University of Liverpool. A section of the medulla columns comprising the four most proximal axonal terminals was used to quantify the number of swellings and regions with disorganised MTs.

Time lapse imaging of cultured primary neurons (in Schneider's/FCS) and fibroblasts (in Gibco's Ham's F-12 medium with 10% FCS) was performed on a Delta Vision Core (Applied Precision) restoration microscope using a [*100x/1.40 UPlan SAPO (Oil)*] objective and the Sedat Quad filter set (*Chroma #89000*). Images were collected using a Coolsnap HQ2 (Photometrics) camera. The temperature was set to 26°C for fly neurons and 37°C for fibroblasts. Time lapse movies were constructed from images taken every 2 s for two mins. To analyse MT dynamics, Eb1::GFP comets were tracked manually using the 'manual tracking' plug-in of ImageJ.

Images were derived from at least two independent experimental repeats performed on different days, for each of which at least three independent culture wells were analysed by taking a minimum of 20 images per well. For statistical analyses, Kruskal–Wallis one-way ANOVA with *post hoc* Dunn's test or Mann–Whitney Rank Sum Tests (indicated as $P_{MW}$) were used to compare groups, and $\chi 2$ tests (indicated as $P_{X2}$) were used to compare percentages. All raw data of our analyses are provided as supplementary Excel/Prism files.

## Molecular biology

EGFP tags are based on *pcDNA3-EGFP* (RRID:Addgene_13031) or *pUAST-EGFP*. All *Drosophila melanogaster efa6* constructs are based on cDNA clone *IP15395* (Uniprot isoform C, intron removed). *Caenorhabditis elegans efa-6* (Y55D9A.1a) constructs are derived from pCZGY1125-efa-6-pcr8 (kindly provided by Andrew Chisholm). *Homo sapiens* PSD1 (ENST00000406432.5, isoform 202) constructs were PCR-amplified from pLC32-hu-psd1-pcr8 vector (kindly provided by Andrew Chisholm). *Homo sapiens* PSD2 (ENST00000274710.3, isoform 201, 771aa) constructs were PCR-amplified from pLC33-hu-psd2-pcr8 vector (kindly provided by Andrew Chisholm). *Homo sapiens* PSD3 was PCR-amplified from pLC34 hu-psd3-pcr8 vector (kindly provided Andrew Chisholm). Note that the PSD3 cDNA clone is most closely related to isoform 201 (ENST00000286485.12: 513aa) and therefore lacks the putative N-terminus found in isoform 202 (ENST00000327040.12). However, the putative MTED core sequence is encoded in the C-terminal PH domain (orange in *Figure 3C*), not the potential N-terminus. *Homo sapiens* PSD4 (ENST00000441564.7, isoform 205) was PCR-amplified from pLC35-hu-psd4-pcr8 vector (kindly provided by Andrew Chisholm). The CAAX motif is derived from human KRAS (V-KI-RAS2 KIRSTEN RAT SARCOMA VIRAL ONCOGENE HOMOLOG). The *Dm*Efa6-NtermΔSxiP::EGFP (aa1-410) insert was synthesised by GeneArt Express (Thermo-Fisher). All construct were cloned using standard (SOE) PCR/ligation based methods, and constructs

and inserts are detailed in Table T1. To generate transgenic fly lines, *P[acman]M-6-attB-UAS-1-3-4* constructs were integrated into *PBac{yellow[+]-attP-3B}VK00031* (Bloomington line #9748; RRID: BDSC_9748) via PhiC31 mediated recombination (outsourced to Bestgene Inc). For details on vectors, sources and inserts see *Table 1*.

## In silico analyses

To generate the phylogenetic tree of Efa6/PSD full length isoforms and N-terms of different species (see *Figure 3—figure supplement 5*), their amino acid sequences were aligned using Muscle or ClustalO (*Goujon et al., 2010*; *McWilliam et al., 2013*; *Sievers et al., 2011*). ProtTest (*Abascal et al., 2005*; *Darriba et al., 2011*) was used to determine amino acid frequencies in the protein datasets and to identify the optimal amino acid substitution model to be used for the Bayesian inference (VT+I+G+F). CUDA-Beagle-optimised MrBayes (*Ronquist et al., 2012*) was run using the VT+I+G+F model [prset statefreqpr = fixed(empirical); lset rates = invgamma] using five chains (one heated) and nine parallel runs until the runs converged and standard deviation of split frequencies were below 0.015 (0.06 for N-terms); PSRF+ was 1.000 and min ESS was >1300 for the TL, alpha and pinvar parameters. The *Drosophila melanogaster* Sec7-PH domain-containing protein Steppke was used as outgroup in the full length tree. Archaeopteryx (*Han and Zmasek, 2009*) was used to depict the MrBayes consensus tree showing branch lengths (amino acid substitutions per site) and Bayesian posterior probabilities.

To **identify a potential MTED in PSD1**, previously identified Efa6 MTED motifs (*O'Rourke et al., 2010*) of 18 orthologues were aligned to derive an amino acid logo. Further orthologues were identified and used to refine the logo. Invariant sites and sites with restricted amino acid substitutions were determined (most prominently MxG-stretch). Stretches containing the invariant MxG stretch were aligned among vertebrate species to identify potential candidates. Berkley's Weblogo server (*Crooks et al., 2004*) was used to generate amino acid sequence logos for each phylum using MTED (ExxxMxGE/D) and MTED-like (MxGE/D) amino acid stretches.

## In vitro analyses

### Protein expression and purification

*Drosophila* Efa6-ΔCterm was cloned into a modified pFastBac vector containing an N-terminal *His6* tag and C-terminal *eGFP* and *StrepII* tags. Recombinant protein was expressed in Sf9 insect cells for 72 hr using a *Baculovirus* system. The protein was purified via a two-step protocol of Ni-affinity using a 1 ml His-Trap column (GE Healthcare) in Ni-affinity buffer [50 mM Tris pH 7.5, 300 mM NaCl, 1 mM Mg C$_{l2}$, 10% (v/v) glycerol] and elution with 200 mM imidazole, followed by Step-tag affinity chromatography using StepTactin resin (GE Healthcare) in BRB20, 75 mM KCl. 0.1% Tween 20, 10% (v/v) glycerol and elution in the same buffer with 5 mM desthiobiotin. MTED peptide (Genscript) was shipped as lyophilised powder with a purity of 95.2%. Upon arrival peptide was dissolved in ultrapure water and used directly.

### MT binding assays

GMPCPP-stabilised, rhodamine-labeled MTs were grown from porcine brain tubulin and adhered to the surface of flow chambers (*Helenius et al., 2006*). 20 nM Efa6-ΔCterm::GFP (in BRB20 pH 6.9, 75 mM KCl, 0.05% Tween20, 0.1 mg/ml BSA, 1% 2-mercaptoethanol, 40 mM glucose, 40 mg/ml glucose oxidase, 16 mg/ml catalase) or 20 nM MCAK::GFP (in the same buffer plus 1 mM ATP and 1 mM taxol) was introduced to the MT-containing channel. Images were recorded using a Zeiss Observer.Z1 microscope equipped with a Zeiss Laser TIRF three module, QuantEM 512SC EMCDD camera (Photometrics) and 100x objective (Zeiss, alphaPlanApo/1.46NA oil). Images of rhodamine-labeled MTs using a lamp as the excitation source and GFP fluorescence using TIRF illumination via a 488 nm laser were collected as described (*Patel et al., 2016*). For both rhodamine and GFP imaging an exposure time of 100 ms was used. The mean GFP intensity on individual MTs was determined from the mean pixel intensity of lines drawn along the long-axis of individual microtubules in Fiji (*Schindelin et al., 2012*). The rhodamine signal was used to locate the position of MTs in the GFP images. Intensity from a region of background was subtracted.

## MT depolymerisation assays

GMPCPP-stabilised, rhodamine-labelled MTs were grown from porcine brain tubulin and adhered to the surface of flow chambers (*Helenius et al., 2006*). Images of a field of fluorescent microtubules were recorded using a Zeiss Observer.Z1 microscope, collecting 1 image every 5 s with an exposure time of 100 ms. Efa6-ΔCterm::GFP (14 nM), MCAK (40 nM) in solution (BRB20 pH 6.9, 75 mM KCl, 1 mM ATP, 0.05% Tween 20, 0.1 mg/ml BSA, 1% 2-mercaptoethanol, 40 mM glucose, 40 mg/ml glucose oxidase, 16 mg/ml catalase) were added to the channel 1 min after acquisition had commenced. Depolymerisation rates were determined from plots of the length of individual microtubules versus time, obtained by thresholding and particle analysis of images using Fiji (*Schindelin et al., 2012*).

## Microtubule growth assays

30 µM 25% rhodamine-labelled porcine brain tubulin was incubated in 80 mM PIPES pH6.9, 5 mM MgCl$_2$, 1 mM EGTA, 5% DMSO and 1 mM GTP at 37°C for 30 min in the presence of either no peptide, 30 µM MTED peptide (APRFEAYMMTGDLILNLSRT; synthesised by Genosphere Biotechnologies Genscript), 30 µM scrambled peptide (MITAPREFDYLNLRAGLSMT; synthesised by Genosphere Biotechnologies) or 8 µM MCAK (1 mM Mg.ATP was included with MCAK). To stabilise MTs and to reduce their density so that they can be easily imaged, the reactions were then diluted 200-fold into BRB80 buffer (80 mM PIPES pH6.9, 1 mM MgCl$_2$, 1 mM EGTA) containing 1 mM taxol. Samples were added to poly-lysine coated cover glasses and imaged by fluorescence microscopy. The amount of tubulin polymer in each field of view was quantified by segmenting the images into background and microtubules by application of a threshold and measuring the total area of tubulin polymer for each image. Five fields of view were quantified from each of two separate experiments.

## Tubulin pull-down assays

MTED peptide or scrambled peptide was coupled (5 ng peptide/µl beads) via the N-terminal amine to cyanogen bromide-activated Sepharose beads (GE Healthcare). 15 µM porcine brain tubulin was incubated with either MTED peptide-coated, scrambled peptide-coated or uncoated sepharose beads in BRB80 with 0.2% Tween 20 for 30 mins at 20°C. The beads were washed five times with a 2:1 *v/v* ratio of BRB80 with 0.2% Tween 20 to beads. An equal volume of 2x Laemmli buffer was added to the washed beads, incubated at 90°C for 5 min, spun down and the supernatant run on a 12% SDS-PAGE gel. The intensity of the bands was quantified in FIJI. Full length human MCAK was coupled (1 µg/ µl beads) to cyanogen bromide-activated Sepharose beads via any solvent exposed lysine or the N-terminal amine. MCAK coated beads were incubated with 5 mM AMPPNP or ADP for 30 min at 20°C and then incubated with 15 µM porcine brain tubulin for 30 mins at 20°C. The beads were then treated as for peptide-coated beads (above) but with inclusion of the appropriate nucleotide in the washing buffer.

## *Xenopus* assays

Interphase cytosol extracts from *Xenopus* eggs were obtained as described previously (*Allan and Vale, 1991*). MT depolymerisation was assessed in a microscopic flow chamber (*Vale and Toyoshima, 1988*) where *Xenopus* cytosol (1 µl cytosol diluted with 20 µl acetate buffer) was incubated for 20 min to allow MTs to polymerise. Then cytosol was exchanged by flow through with Efa6-ΔCterm::GFP, MCAK or synthetic MTED peptide (all 20 nM in acetate buffer pH 7.4: 100 mM K-Acetate, 3 mM Mg-Acetate, 5 mM EGTA, 10 mM HEPES), and MT length changes observed by recording 10 random fields via VE-DIC microscopy (*Allan, 1993*; *Allan and Vale, 1991*). MT polymerisation was analysed in a microscope flow cell containing 9 µl diluted *Xenopus* egg cytosol (see above) to which 1 µl acetate buffer was added, either alone or containing 20 nM MTED. After 10 min, 20 random fields were recorded via VE-DIC microscopy for each condition and the numbers of MTs per field counted.

For the in vivo assay, *Xenopus* embryos were injected in one blastomere at the 4 cell stage with 200 ng of mRNA encoding Efa6-Nterm::GFP or mCherry alone. The embryos were imaged at stage 10.25 (*Heasman, 2006*) with a Leica fluorescent stereoscope.

## Acknowledgements

This work was made possible through support by the BBSRC to AP (BB/I002448/1, BB/P020151/1, BB/L000717/1, BB/M007553/1) to NSS (BB/M007456/1) and KD (BB/J005983/1), by parents as well as the Faculty of Life Sciences to YQ, by the Leverhulme Trust to IH (ECF-2017–247) and by the German Research Council (DFG) to AV (VO 2071/1-1). The Manchester Bioimaging Facility microscopes used in this study were purchased with grants from the BBSRC, The Wellcome Trust and The University of Manchester Strategic Fund. The Fly Facility has been supported by funds from The University of Manchester and the Wellcome Trust (087742/Z/08/Z). We thank Tom Millard and Marvin Bentley for very helpful comments on the manuscript, Simon Lowell for advice on the phylogenetic analyses, Hiro Ohkura for kindly providing DmEb1 antibody and Andrew Chisholm for the *C.elegans* Efa6 and human PSD constructs. Stocks obtained from the Bloomington *Drosophila* Stock Center (NIH P40OD018537) were used in this study.

## Additional information

### Funding

| Funder | Grant reference number | Author |
|---|---|---|
| Biotechnology and Biological Sciences Research Council | BB/I002448/1 | Andreas Prokop |
| Biotechnology and Biological Sciences Research Council | BB/P020151/1 | Andreas Prokop |
| Biotechnology and Biological Sciences Research Council | BB/L000717/1 | Andreas Prokop |
| Biotechnology and Biological Sciences Research Council | BB/M007553/1 | Andreas Prokop |
| Biotechnology and Biological Sciences Research Council | BB/M007456/1 | Natalia Sanchez-Soriano |
| Biotechnology and Biological Sciences Research Council | BB/J005983/1 | Karel Dorey |
| Leverhulme Trust | ECF-2017-247 | Ines Hahn |
| Deutsche Forschungsgemeinschaft | VO 2071/1-1 | André Voelzmann |
| BBSRC | BB/R018960/1 | Natalia Sanchez-Soriano |

The funders had no role in study design, data collection and interpretation, or the decision to submit the work for publication.

### Author contributions

Yue Qu, Conceptualization, Data curation, Formal analysis, Validation, Investigation, Visualization, Methodology, Writing—original draft, Project administration; Ines Hahn, Natalia Sanchez-Soriano, Conceptualization, Data curation, Formal analysis, Supervision, Funding acquisition, Validation, Investigation, Visualization, Methodology, Writing—original draft, Project administration, Writing—review and editing; Meredith Lees, Jill Parkin, Investigation, Methodology; André Voelzmann, Data curation, Formal analysis, Funding acquisition, Validation, Investigation, Visualization, Methodology, Writing—original draft, Writing—review and editing; Karel Dorey, Resources, Funding acquisition, Investigation, Methodology, Writing—review and editing; Alex Rathbone, Resources, Formal analysis, Validation, Investigation, Visualization, Methodology; Claire T Friel, Resources, Data curation, Formal analysis, Validation, Investigation, Visualization, Methodology, Writing—review and editing; Victoria J Allan, Resources, Investigation, Methodology; Pilar Okenve-Ramos, Formal analysis, Validation, Investigation, Visualization, Methodology; Andreas Prokop, Conceptualization, Data curation, Supervision, Funding acquisition, Visualization, Methodology, Writing—original draft, Project administration, Writing—review and editing

## Author ORCIDs

Ines Hahn ⓘ https://orcid.org/0000-0001-7703-8160
André Voelzmann ⓘ http://orcid.org/0000-0002-7682-5637
Karel Dorey ⓘ https://orcid.org/0000-0003-0846-5286
Claire T Friel ⓘ https://orcid.org/0000-0001-8395-5301
Victoria J Allan ⓘ https://orcid.org/0000-0003-4583-0836
Pilar Okenve-Ramos ⓘ http://orcid.org/0000-0002-7513-6557
Natalia Sanchez-Soriano ⓘ https://orcid.org/0000-0002-6667-2817
Andreas Prokop ⓘ http://orcid.org/0000-0001-8482-3298

## Ethics

Animal experimentation: All experiments involving Xenopus laevis were approved by the Ethical Review Committe of the University of Manchester and a Home Office license (ref. PFDA14F2D).

## Decision letter and Author response

Decision letter https://doi.org/10.7554/eLife.50319.sa1
Author response https://doi.org/10.7554/eLife.50319.sa2

# Additional files

## Supplementary files

• Transparent reporting form

## Data availability

All data generated or analysed during this study are included in the manuscript and supporting files. Source data files have been provided.

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
