## [Decision Letter]

**Decision letter after peer review:**

[Editors’ note: a previous version of this study was rejected after peer review, but the authors submitted for reconsideration. Following rejection after re-review, the authors then appealed the decision and the appeal was accepted. The first decision letter after peer review is shown below.]

Thank you for submitting your work entitled "Efa6 regulates axon growth, branching and maintenance by eliminating off-track microtubules at the cortex" for consideration by *eLife*. Your article has been reviewed by a Senior Editor, a Reviewing Editor, and three reviewers. The three reviewers have opted to remain anonymous.

Our decision has been reached after consultation between the reviewers. Based on these discussions and the individual reviews below, we regret to inform you that your work will not be considered further for publication in *eLife*.

All three referees agreed your work represents a thorough and detailed characterization of the function of Efa6 in axon growth and the potential roles in the microtubule cytoskeleton. However, as you will see in the reviews, and in the following discussions, the referees agreed that the advance over previous work was minimal. The O'Rourke and Chen studies referenced by two referees appears to undermine the novelty of the work as well. In light of these and other considerations based on the referee comments, we are unable to pursue your manuscript further.

Reviewer #1:

In this manuscript the authors show that the exchange factor for Arf6 (EFa6) functions in depolymerizing microtubules (MTs) in axons to keep MTs bundled in the axon shaft. This is a well-documented and quantified study that adds to our knowledge of this protein in nervous system function. Although the authors show that the full-length protein and the MT-eliminating domain (MTED) serve to depolymerize MTs in the axon and growth cone, they are not able to discern the molecular mechanism by which this protein and domain function in MT elimination. In *C. elegans*, previous studies have shown that the Efa6 protein and the MTED domain function in limiting MT growth at the cell cortex in dividing cells (O'Rourke et al., 2010) and that Efa6 functions in axon regeneration through interactions with MT-associated proteins Tac-1 and Zyg-8 (doublecortin-like kinase) (Chen et al., 2011; 2015). Thus, this study adds to the literature by showing Efa6 and its MTED also function in axon outgrowth, branch formation and maintenance in a similar manner and provide evidence that Efa6 cooperates with Shot protein to accomplish these tasks. Shot appears to act, shown through previous studies by this lab, as a MT bundle organizer and Efa6 is proposed to work as a quality control mechanism to keep MTs from going off track. The major concern of this reviewer is how much this study really adds to our understanding of Efa6 function in the nervous system. The protein was already known to limit MT polymerization at the plasma membrane and the MTED was previously shown to be critical for this function, both in non-neuronal cells and neurons. Yes, this work adds to the literature by showing Efa6 also acts in axon outgrowth, branching and maintenance but the authors are not able to determine the mechanism of action. Thus, this seems to be an incremental advance.

Reviewer #2:

Qu et al. investigate *Drosophila* Efa6 in neurons and show that reduction in Efa6 results in an increase in axon length and more branches, while overexpression of Efa6 has the opposite phenotype. Further, Efa6 reduction results in more axonal swellings and some of these contain disorganised microtubules. Overexpression of Efa6 or of N-terminal fragments containing the so-called microtubule elimination domain (MTED) in fibroblasts reduces MTs at the cell cortex. These data are nice and convincing, but in line with previous findings in *C. elegans* (O'Rourke et al., 2010 and Chen et al., 2015). Thus, the study largely corroborates previous findings. Unfortunately, a conceptual advance how Efa6 functions is lacking, mainly because the attempt to reconstitute the proposed process of cortex-induced catastrophe with purified proteins failed. This is probably not surprising as interacting partners of the 18 amino acid MTED have already been identified by others (Chen et al., 2015), which could mediate microtubule-regulating functions. While it might not be absolutely necessary to reconstitute the effect in vitro, the cell biology experiments provided are also lacking mechanistic insight as most only show a snapshot of defects after several days of inhibition/overexpression. The high-resolution imaging possible in the fibroblasts experiments has not been employed to study the interactions of Efa6 with microtubules at the cell cortex and thus to provide direct evidence that and new insight how Efa6 eliminates off-track microtubules.

Essential revisions:

1) It is unclear why the authors used lower amounts of Efa6 in their in vitro assays than MCAK even though they show that Efa6 binds weaker to microtubules than MCAK. Also, why would the authors assume that Efa6 could depolymerise GMPCPP-stabilised microtubules? Surely one would try to see an effect on dynamic microtubules first. If the proposal is that Efa6 triggers catastrophe at the cell cortex, then slowing down assembly is all what it takes to have the desired effect. Further, it seems dangerous to lyophilise and then reconstitute purified Efa6 in water. While this might be appropriate for some proteins, I wouldn't want to conclude that a protein has no effect on microtubule dynamics unless I used at least once a fresh prep of protein. Thus, I am not convinced that we can take the manuscript as negative evidence for a direct function of Efa6 on microtubules.

2) The authors propose that shot causes more off track MTs and Efa6 removes them. This is testable by expressing Efa6 in shot mutants, which should correct the defects. Shot mutants should increase the incidents of off track MTs whose interactions with Efa6 could then be studied more easily, ideally by recruiting Efa6 locally and determining the cortex dwell times of off track MTs in the Efa6 positive regions relative to normal or depleted cell cortex.

3) The number of EBs and filopodia per neuron or growth cone will depend at least partially on its size. Thus as neurons are small and don't have a proper growth cone when Efa6 is overexpressed (Figure 6G,H), it is expected that the numbers of other items are also reduced and vice versa in Efa6 mutants. The data in Figure 6 and Figure 7 need to be shown relative to neuron size / axon length / growth cone (surface) area. The distributions should be shown also for the control and actual values presented. For example, lifetime should be given in seconds and speed in nm/s, counts as a frequency per µm of axon or µm^2^ of growth cone. The authors will need to reassess whether statistically significant effects are present once the data have been normalised correctly.

4) Distributions or error bars are missing for a number of panels. Count data with two categories do not need to be shown as stacked bars, but should rather contain error bars or the exact values for technical repeats / independent experiments. Some data are just mentioned in the text. In particular, dwell times of microtubule tips at the cell cortex are key given the title of the manuscript, but no quantitative data are shown in the figures.

Reviewer #3:

This work is a comprehensive study on the function of efa6 in regulating axonal growth and axonal maintenance. It uses a combination of systems to conclude that *Drosophila* (and *C. elegans*, together with published work) efa6 may be a cortical MT collapse factor eliminating MTs that have left the axon bundle (but that its function is not conserved in vertebrates).

The phenotypes regarding axon length, and MT organization and/or levels are quite striking, both in flies and in fibroblasts. However, it is a pity that the mechanism of action is not clarified, especially since MTED had already been described to have such a function in *C. elegans*. I am not sure the data has eliminated the possibility of efa6 having an effect on the biophysics of the membrane/cortex, and this effect indirectly impacting on MTs. Could the authors discuss this?

Can the authors explain why they analyzed efa6 in the first place? It is not described in the Introduction what was the argument for this study.

An extremely dense paper, where it feels on occasions that the authors want to present everything they have tested, which makes some parts of the manuscript difficult to read. I would suggest the shortening/reduction of some sections. For example, the domain characterization could start with MTED, since this domain has already been shown to have a MT destabilizing function in *C. elegans*, and the *C. elegans* efa6 has no PDZ domain, for example.

The domain analysis was done in the presence of endogenous protein, so it is harder to conclude what the primary function of the domain is, as the phenotypes can result from complex dominant effects, or they may rely on the endogenous full length. For the key domains, would it be possible to test the effect in the absence of endogenous prot? Maybe transfecting primary cultured neurons from the mutant flies? For example, what happens when the only efa6 protein expressed in cells is a full length one with a mutated MTED?

[Editors’ note: what now follows is the decision letter after the authors submitted for further consideration.]

Thank you for choosing to send your work entitled "Efa6 protects axons and regulates their growth and branching by inhibiting microtubule polymerisation at the cortex" for consideration at *eLife*. Your article and your letter of appeal and response to referee comments have been considered by a Senior Editor, a reviewing editor, and three referees of your original submission, and we regret to inform you that we are upholding our original decision.

Please note that the referees believe your work is improved, but is not sufficiently rigorous in many instances due to missing controls and analysis (noted by the referees in the comments below) to provide mechanistic Efa6 functions in microtubule regulation. There have also been a number of issues raised about the writing and more importantly referencing key papers in the field.

*Reviewer #1:*

I think that the paper has improved significantly mainly by shortening the writing, however, major issues remain and I don't feel that my previous comments have been adequately addressed in the revised version.

Note: As I feel that reviewers should not be expected to download and consider data from an author's private website, I did not take into consideration the source data tables nor the supplementary movies that were not uploaded together with the manuscript. However, I don't expect that their availability would have changed my conclusions significantly.

The authors now claim to have directly shown that MTED blocks MT growth in vitro and in cells, thereby providing new mechanistic insight. The data that supposedly show this are provided in a minimalistic Figure 4 and a supplementary movie which wasn't available on the journal's website.

The "successful" in vitro experiments were done with equimolar and a tenfold excess of MTED peptide to tubulin. It is highly unlikely that 30 – 300 μm MTED is physiologically relevant and the observed effect might not even be specific. An adequate control for this experiment would be to show that a scrambled peptide does not have a similar effect at this concentration. As the authors have a Efa6::GFP knock-in line, it should be possible to determine the endogenous concentration of Efa6 relative to a GFP standard and then repeat in vitro experiments at physiologically relevant peptide concentrations including an adequate control. Furthermore, the results should be quantified in order to sustain the claim in subsection “The MTED is a good predictor of MT-inhibiting function directly affecting MT polymerisation”: "resulted in strong suppression of MT polymerisation in a dose-dependent manner". The figure only shows a single snapshot of Taxol-stabilised MTs. Polymerisation itself has not been measured or visualised for either concentration of MTED peptide. In light of the data supporting a direct role of Efa6 MTED towards microtubules being so weak, I repeat that it would be appropriate to discuss previously published findings of Efa6 interactors that are known to interact with microtubules and might work together with Efa6 to affect microtubule assembly in the vicinity of the cell cortex. Given the SxIP related motifs in Efa6, EBs are additional potential interactors that have been implicated in regulating microtubule dynamics at the cell cortex before.

The requested data describing microtubule behaviour in Efa6-expressing fibroblasts were not provided beyond a statement referring to a supplementary movie. As those are key to the mechanistic understanding of Efa6 action and for supporting a key claim in the manuscript title, data need to be provided in a main figure or table and include a robust quantification of the phenomenon.

Therefore, the manuscript still falls short in providing convincing evidence for Efa6 to inhibit microtubule polymerisation at the cell cortex.

There are several incidences of overstatement / oversimplification, e.g.:

1) The data in Figure 3B show that overexpression of Efa6-Nterm-^ΔMTED^ has a significant effect on axon loss. This is weaker than that of Efa6-Nterm or of Efa6-FL, but it is still statistically significant, suggesting that there is residual activity in that fragment. Likewise, in fibroblasts, Efa6-Nterm-^ΔMTED^ results in complete loss of microtubules in a few cells, which did not happen in the GFP control. However, the authors state in subsection “The N-terminal 18aa motif of Efa6 is essential for microtubule-436 inhibiting activity of Efa6”: "Efa6-Nterm-^ΔMTED^::GFP behaved like GFP controls". The statements should be corrected and any additional activity towards microtubules outside of the MTED region should be discussed.

2) The data in Figure 3C-D show that human full length PSD1-3 have a significant effect on microtubules (p=0.001!!!), but the authors state in subsection “The MTED is a good predictor of MT-inhibiting function directly affecting MT polymerisation”: "none of the 6 human constructs.… showed MT collapse (Figure 3C-E)". Actually, the effect of the full-length PSD constructs cannot be matched by the N-terminal portion of the molecules or the MTED-like region in the human proteins. This discrepancy should be described and discussed rather than hidden in the paper.

Comments on writing and data presentation:

1) The paper suffers from an inadequately high rate of self-citations, while lacking balanced representation of other work in the field. One example is the citing of two review articles and a preprint involving the senior author for the statement in the Introduction: "Here we make use of *Drosophila* neurons as a well-established, powerful model for studying roles of MT regulators". This statement should instead be followed by references to landmark papers in the field in which key discoveries were first described.

2) The figures don't align with the text. Some panels are never cited and those that are cited are in random order, for example Figure S3 (now Figure 3—figure supplement 1) and Figure S5 (now Figure 3—figure supplement 2)are cited well before Figure S1 (now Figure 3—figure supplement 5) and Figure S2 (now Figure 3—figure supplement 4) are mentioned. Figure 2D is cited before Figure 1A-C or 2A-C were mentioned etc.

3) The figure legends lack structure and information on the number of independent experiments from which the data were obtained – the methods section doesn't provide this information either for the majority of the experiments. Any raw data to be considered should be uploaded with the manuscript as supplementary information. The authors need to state for each panel what the sample number refers to: individual measurements, cells/embryos analysed or independent experimental repeats.

4) There is unnecessary reference to methods and papers where methods have been used before in the Results section … and specific bar colours within the main text (e.g. Results section and Discussion section), reducing readability.

5) The methods section does not contain sufficient information without needing to refer to a whole lot of the lab's previous papers. For example, in subsection “*Drosophila* primary cell culture”: "per 100 μl dispersion medium (Prokop et al., 2012)" even the composition of media is not included.

6) The authors should explain the differences between the various Efa6 mutant strains used and a rationale why certain experiments were done with only one particular strain. For example, why is EB1 comet lifetime in filopodia shown for GX6[w+], but in growth cones for GX6[w-]? Why are filopodia per growth cone with Eb1 comets shown for GX6[w-] and KO#1, but filopodia per growth cone with MTs for GX6[w-] and GX6[w+], but not for the KO line? One can't help to suspect that data that didn't fit the scheme were excluded. If this isn't the case, the authors should provide an explanation for the inconsistencies.

*Reviewer #2:*

In my previous review my main concern was how much this study really adds to our understanding of Efa6 function in the nervous system. The protein was already known to limit MT polymerization at the plasma membrane and the microtubule elimination domain (MTED) was previously shown to be critical for this function, both in non-neuronal cells and neurons. The authors have now added new data (Figures 4, Figure 6G and Figure 8N). Importantly, Figure 4 shows that the MTED peptide is capable of limiting MT polymerization in vitro when only tubulin and MTED peptide is present. Furthermore, they show that the MTED peptide is able to immunoprecipitate purified tubulin. These experiments suggest that the MTED can function directly with tubulin. These data do provide some mechanism that was lacking in the previous version of the manuscript. However, it is somewhat crude to compare MTED inhibition of polymerization to no addition of MTED (in the growth assay in Figure 4A) or by comparing MTED-coupled beads to empty beads (in the IP in Figure 4B). These are very weak negative controls, given that tubulin readily interacts non-specifically with other proteins in many assays and there are not any positive controls in the assays they perform in Figure 4. In Figure 4—figure supplement 1A and B they use MCAK in their in vitro assays as a positive control. Since the Efa6-δ-Cterminal peptide does not seem to work in Figure 4—figure supplement 1A/B (the authors suggest this may be due to large disordered regions in that part of the protein), some other (scrambled?) peptide could be used as a negative control and/or MCAK (or peptide thereof) as a positive control. Such experiments would provide a more compelling mechanism with relatively little additional work. Additionally, the authors have also modified the previous manuscript to make it more coherent and compact and addressed my previous minor concerns.

*Reviewer #3:*

This work is a comprehensive study on the function and molecular mechanism of efa6 in regulating MT organization during axonal growth and axonal maintenance. It uses a combination of systems to conclude that *Drosophila* efa6 is a cortical MT collapse factor blocking growth of MTs that have left the axon bundle (but that its function is not conserved in vertebrates). The authors have answered most of the points I raised in a previous review, and the manuscript has improved greatly, especially regarding the possible mechanism of efa6 action.

The only comment is that the order of Figures is slightly mixed, as in it would be better to start the results with Figure 1A, instead of Figure 1F, and the first figure supplement I encountered was Figure 3—figure supplement 1B, instead of Figure S1 (now Figure 3—figure supplement 5. In addition, there are various typos and unclear sentences, e.g., subsection “The MTED is a good predictor of MT-inhibiting function directly affecting MT polymerisation”.

---

## [Author Response]

[Editors’ note: the author responses to the first round of peer review follow.]

All three referees agreed your work represents a thorough and detailed characterization of the function of Efa6 in axon growth and the potential roles in the microtubule cytoskeleton. However, as you will see in the reviews, and in the following discussions, the referees agreed that the advance over previous work was minimal. The O'Rourke and Chen studies referenced by two referees appears to undermine the novelty of the work as well. In light of these and other considerations based on the referee comments, we are unable to pursue your manuscript further.Reviewer #2:Qu et al. investigate Drosophila Efa6 in neurons and show that reduction in Efa6 results in an increase in axon length and more branches, while overexpression of Efa6 has the opposite phenotype. Further, Efa6 reduction results in more axonal swellings and some of these contain disorganised microtubules. Overexpression of Efa6 or of N-terminal fragments containing the so-called microtubule elimination domain (MTED) in fibroblasts reduces MTs at the cell cortex. These data are nice and convincing, but in line with previous findings in C. elegans (O'Rourke et al., 2010 and Chen et al., 2015). Thus, the study largely corroborates previous findings. Unfortunately, a conceptual advance how Efa6 functions is lacking, mainly because the attempt to reconstitute the proposed process of cortex-induced catastrophe with purified proteins failed. This is probably not surprising as interacting partners of the 18 amino acid MTED have already been identified by others (Chen et al., 2015), which could mediate microtubule-regulating functions. While it might not be absolutely necessary to reconstitute the effect in vitro, the cell biology experiments provided are also lacking mechanistic insight as most only show a snapshot of defects after several days of inhibition/overexpression. The high-resolution imaging possible in the fibroblasts experiments has not been employed to study the interactions of Efa6 with microtubules at the cell cortex and thus to provide direct evidence that and new insight how Efa6 eliminates off-track microtubules.Essential revisions:1) It is unclear why the authors used lower amounts of Efa6 in their in vitro assays than MCAK even though they show that Efa6 binds weaker to microtubules than MCAK. Also, why would the authors assume that Efa6 could depolymerise GMPCPP-stabilised microtubules? Surely one would try to see an effect on dynamic microtubules first. If the proposal is that Efa6 triggers catastrophe at the cell cortex, then slowing down assembly is all what it takes to have the desired effect. Further, it seems dangerous to lyophilise and then reconstitute purified Efa6 in water. While this might be appropriate for some proteins, I wouldn't want to conclude that a protein has no effect on microtubule dynamics unless I used at least once a fresh prep of protein. Thus, I am not convinced that we can take the manuscript as negative evidence for a direct function of Efa6 on microtubules.

This slowing effect is exactly what we observe in our new fibroblast experiments (Video 2 and Video 2 which can be accessed here: http://www.prokop.co.uk/Qu+al/SupplMov.html).

2) The authors propose that shot causes more off track MTs and Efa6 removes them. This is testable by expressing Efa6 in shot mutants, which should correct the defects. Shot mutants should increase the incidents of off track MTs whose interactions with Efa6 could then be studied more easily, ideally by recruiting Efa6 locally and determining the cortex dwell times of off track MTs in the Efa6 positive regions relative to normal or depleted cell cortex.

We have added data where we overexpress Efa6 in *shot* mutant neurons and find a strong improvement of the MT disorganisation phenotype, as predicted by the reviewer (Figure 6G and Figure 8N).

3) The number of EBs and filopodia per neuron or growth cone will depend at least partially on its size. Thus as neurons are small and don't have a proper growth cone when Efa6 is overexpressed (Figure 6G,H), it is expected that the numbers of other items are also reduced and vice versa in Efa6 mutants. The data in Figure 6 and Figure 7 need to be shown relative to neuron size / axon length / growth cone (surface) area. The distributions should be shown also for the control and actual values presented. For example, lifetime should be given in seconds and speed in nm/s, counts as a frequency per µm of axon or µm^2^ of growth cone. The authors will need to reassess whether statistically significant effects are present once the data have been normalised correctly.

As stated in the text, filopodia numbers are unchanged in growth cones of Efa6, as compared to wild-type controls. Sizes of growth cones are therefore negligible. The number of EB1 comets and filopodia per neuron was from start assessed relative to axon length but not properly indicated. This has now been corrected in the Figure titles of Figure 6 (previous Figure 7).

Eb1 comet life-time in seconds and velocity in µm/s as reference data for controls were added to the Figure legends of Figure 5 and Figure 6.

4) Distributions or error bars are missing for a number of panels. Count data with two categories do not need to be shown as stacked bars, but should rather contain error bars or the exact values for technical repeats / independent experiments. Some data are just mentioned in the text. In particular, dwell times of microtubule tips at the cell cortex are key given the title of the manuscript, but no quantitative data are shown in the figures.

We followed common procedure for all categorised data which were assessed with Chi^2^ tests. Since this is common practise and many data sets contain three categories where data need to be shown as stacked bars, we had originally taken the decision to assess all categorised data sets the same way and do not feel that changing this strategy would improve the outcome, hence justify the enormous effort of re-assessing a large data set.

Quantitive data for dwell times in filopodia are shown in Figure 5M and precise data also mentioned in the text.

Reviewer #3:This work is a comprehensive study on the function of efa6 in regulating axonal growth and axonal maintenance. It uses a combination of systems to conclude that Drosophila (and C. elegans, together with published work) efa6 may be a cortical MT collapse factor eliminating MTs that have left the axon bundle (but that its function is not conserved in vertebrates).The phenotypes regarding axon length, and MT organization and/or levels are quite striking, both in flies and in fibroblasts. However, it is a pity that the mechanism of action is not clarified, especially since MTED had already been described to have such a function in C. elegans. I am not sure the data has eliminated the possibility of efa6 having an effect on the biophysics of the membrane/cortex, and this effect indirectly impacting on MTs. Could the authors discuss this?

We now show in vitro data that Efa6 binds tubulin and interferes with polymerisation (new Figure 4).

Can the authors explain why they analyzed efa6 in the first place? It is not described in the Introduction what was the argument for this study.

The whole rationale of the work has been re-written, using as an angle the lack of understanding of observed axon growth phenotypes upon functional manipulation of cortical collapse factors in *C. elegans* (Efa6) and mammals (Kif21B).

An extremely dense paper, where it feels on occasions that the authors want to present everything they have tested, which makes some parts of the manuscript difficult to read. I would suggest the shortening/reduction of some sections. For example, the domain characterization could start with MTED, since this domain has already been shown to have a MT destabilizing function in C. elegans, and the C. elegans efa6 has no PDZ domain, for example.

The paper has been completely re-written and shortened to a great deal.

The domain analysis was done in the presence of endogenous protein, so it is harder to conclude what the primary function of the domain is, as the phenotypes can result from complex dominant effects, or they may rely on the endogenous full length. For the key domains, would it be possible to test the effect in the absence of endogenous prot? Maybe transfecting primary cultured neurons from the mutant flies? For example, what happens when the only efa6 protein expressed in cells is a full length one with a mutated MTED?

We do not see what would be gained from this experiment: (1) we now know that the MTED clearly affects MT polymerisation in vitro in the absence of other protein; (2) the data obtained from our comprehensive series of experiments in fibroblasts (which do not harbour any MT-eliminating endogenous Efa6) draw a very clear picture and are fully consistent with our findings in neurons.

[Editors' note: the author responses to the re-review follow.]

Thank you for choosing to send your work entitled "Efa6 protects axons and regulates their growth and branching by inhibiting microtubule polymerisation at the cortex" for consideration at eLife. Your article and your letter of appeal and response to referee comments have been considered by a Senior Editor, a reviewing editor, and three referees of your original submission, and we regret to inform you that we are upholding our original decision.Please note that the referees believe your work is improved, but is not sufficiently rigorous in many instances due to missing controls and analysis (noted by the referees in the comments below) to provide mechanistic Efa6 functions in microtubule regulation. There have also been a number of issues raised about the writing and more importantly referencing key papers in the field.Reviewer #1:An adequate control for this experiment would be to show that a scrambled peptide does not have a similar effect at this concentration.

These experiments together with MCAK as a positive control are now provided and shown in Figure 4. These data clearly show that scrambled peptide does not inhibit microtubule growth, whereas MTED and MCAK inhibit growth of microtubules.

As the authors have a Efa6::GFP knock-in line, it should be possible to determine the endogenous concentration of Efa6 relative to a GFP standard and then repeat in vitro experiments at physiologically relevant peptide concentrations including an adequate control.

Since Efa6 is compartmentalised to the membrane, it is close to impossible to deduce any concentration statements from whole cell extracts. Furthermore, it needs to be considered that in vitroconditions will hardly match the cell conditions, and mechanisms might not work as well as in the cellular context. However, the live movies provided (Video 1, Video 2 and Video 4) make a very powerful complementary statement that matches the qualitative statement of our in vitroexperiments and demonstrate their functional relevance.

Furthermore, the results should be quantified in order to sustain the claim in subsection “The MTED is a good predictor of MT-inhibiting function directly affecting MT polymerisation”: "resulted in strong suppression of MT polymerisation in a dose-dependent manner". The figure only shows a single snapshot of Taxol-stabilised MTs. Polymerisation itself has not been measured or visualised for either concentration of MTED peptide.

Quantifications are now provided in Figure 4, and the dose-dependence statement was removed.

In light of the data supporting a direct role of Efa6 MTED towards microtubules being so weak, I repeat that it would be appropriate to discuss previously published findings of Efa6 interactors that are known to interact with microtubules and might work together with Efa6 to affect microtubule assembly in the vicinity of the cell cortex.

Efa6 interactors, in particular of the MTED, are now discussed in great detail in the second half of the second Discussion section. However, note that these interactors

do not contribute to Efa6 function, but carry out a delayed role in growth promotion, i.e. a role opposite to that of Efa6 (which is why this information had not been provided in earlier manuscript versions).

Given the SxIP related motifs in Efa6, EBs are additional potential interactors that have been implicated in regulating microtubule dynamics at the cell cortex before.

Our detailed structure function analyses already included removal of the SxIP motifs which was clearly described in the text and quantified in Figure 3B. We found that abolishing these EB1-interaction sites does not cause a measurable reduction in MT-inhibiting activity in neurons or fibroblasts.

The requested data describing microtubule behaviour in Efa6-expressing fibroblasts were not provided beyond a statement referring to a supplementary movie. As those are key to the mechanistic understanding of Efa6 action and for supporting a key claim in the manuscript title, data need to be provided in a main figure or table and include a robust quantification of the phenomenon.

Robust quantifications are now provided in Figure 4.

Therefore, the manuscript still falls short in providing convincing evidence for Efa6 to inhibit microtubule polymerisation at the cell cortex.There are several incidences of overstatement / oversimplification, e.g.:1) The data in Figure 3B show that overexpression of Efa6-Nterm-^ΔMTED^ has a significant effect on axon loss. This is weaker than that of Efa6-Nterm or of Efa6-FL, but it is still statistically significant, suggesting that there is residual activity in that fragment. Likewise, in fibroblasts, Efa6-Nterm-^ΔMTED^ results in complete loss of microtubules in a few cells, which did not happen in the GFP control. However, the authors state in subsection “The N-terminal 18aa motif of Efa6 is essential for microtubule-436 inhibiting activity of Efa6”: "Efa6-Nterm-^ΔMTED^::GFP behaved like GFP controls". The statements should be corrected and any additional activity towards microtubules outside of the MTED region should be discussed.

The statement has been improved to reflect our data accurately: "only constructs containing the MTED (Efa6-ΔCterm::GFP, Efa6-Nterm::GFP and Efa6-MTED::GFP) caused strong axon loss in neurons and MT network depletion in fibroblasts, whereas Efa6-Nterm^ΔMTED^::GFP caused no phenotypes in fibroblasts and very mild axon loss in neurons (Figures 3B; Figure 3—figure supplement 1C,D,F,G; Figure 3—figure supplement 6C,D,G,H)." From this statement we lead over to assessing the SxIP motif as a potential second motif in the N-terminus (see also two points above).

2) The data in Figure 3C-D show that human full length PSD1-3 have a significant effect on microtubules (p=0.001!!!), but the authors state in subsection “The MTED is a good predictor of MT-inhibiting function directly affecting MT polymerisation”: "none of the 6 human constructs.… showed MT collapse (Figure 3C-E)". Actually, the effect of the full-length PSD constructs cannot be matched by the N-terminal portion of the molecules or the MTED-like region in the human proteins. This discrepancy should be described and discussed rather than hidden in the paper.

This statement has been moderated to accurately reflect our data: “When transfected into fibroblasts, we found that all 6 fly/worm constructs had strong MT-inhibiting properties, whereas the 6 human constructs (PSD1-4 full length, PSD1-Nterm, PSD1-MTED-like) showed only a slight increase in MT network defects that were far from the strong MT depletion observed with the fly/worm constructs; complete depletion of MT networks was never observed with the human constructs (Figure 3C-E).”

Comments on writing and data presentation:1) The paper suffers from an inadequately high rate of self-citations, while lacking balanced representation of other work in the field. One example is the citing of two review articles and a preprint involving the senior author for the statement in the Introduction: "Here we make use of Drosophila neurons as a well-established, powerful model for studying roles of MT regulators". This statement should instead be followed by references to landmark papers in the field in which key discoveries were first described.

We carefully read through the paper under this aspect and do not agree with the reviewer. Since there are currently very few other groups that focus their research on the wider field of axonal MTs in *Drosophila*, it seems natural that our own papers rank higher in proportion. It is particularly difficult to find reviews that cover this particular topic, but we tried our best to find a better balance.

2) The figures don't align with the text. Some panels are never cited and those that are cited are in random order, for example Figure S3 (now Figure 3—figure supplement 1) and Figure S5 (now Figure 3—figure supplement 2)are cited well before Figure S1 (now Figure 3—figure supplement 5) and Figure S2 (now Figure 3—figure supplement 4) are mentioned. Figure 2D is cited before Figure 1A-C or 2A-C were mentioned etc.

We took care to make sure that now all panels are mentioned and that all figures occur in the order as they are mentioned in the main text.

3) The figure legends lack structure and information on the number of independent experiments from which the data were obtained – the methods section doesn't provide this information either for the majority of the experiments. Any raw data to be considered should be uploaded with the manuscript as supplementary information. The authors need to state for each panel what the sample number refers to: individual measurements, cells/embryos analysed or independent experimental repeats.

We re-read the figure legends and made some improvements. The original statement in the methods part: "Images were derived from at least 2 independent experimental repeats performed on different days, for each of which at least 3 independent culture wells were analysed by taking a minimum of 20 images per well." has now been moved into a more prominent location together with statistics explanations, and also been mentioned in the figure legends for clarity. We added to the figure legends what entities were used to indicate sample numbers.

4) There is unnecessary reference to methods and papers where methods have been used before in the Results section and specific bar colours within the main text (e.g. Results section and Discussion section), reducing readability.

In a few positions we removed "see methods", but left it where the methods part explains complex procedures that are kept concise in the Results section (e.g. the Gal80 approach). We disagree with deleting bar colour information from the text, since they directly point the reader to the relevant information in mentioned figures.

5) The methods section does not contain sufficient information without needing to refer to a whole lot of the lab's previous papers. For example, in subsection “Drosophila primary cell culture”: "per 100 μl dispersion medium (Prokop et al., 2012)" even the composition of media is not included.

We re-read the methods part in its entirety and strongly disagree. The amount of detail provided is of high standard. We have added the details for the culture medium.

6) The authors should explain the differences between the various Efa6 mutant strains used and a rationale why certain experiments were done with only one particular strain. For example, why is EB1 comet lifetime in filopodia shown for GX6[w+], but in growth cones for GX6[w-]? Why are filopodia per growth cone with Eb1 comets shown for GX6[w-] and KO#1, but filopodia per growth cone with MTs for GX6[w-] and GX6[w+], but not for the KO line? One can't help to suspect that data that didn't fit the scheme were excluded. If this isn't the case, the authors should provide an explanation for the inconsistencies.

Comment has been inserted: "Efa6 knock-down (Efa6-RNAi), two overlapping deficiencies uncovering the entire Efa6 gene locus (Efa6Def), and three precise gene deletions including Efa6^GX6[w+]^, Efa6^GX6[w-]^ and Efa6^KO#1^ (see methods for details; Huang et al., 2009). In all these conditions, axon length at 6 HIV was increased compared to wild-type by at least 20% (Figure 2A,B,D). Since there were no obvious differences between the precise deletion lines, these alleles were used interchangeably for further experiments."

Reviewer #2:In my previous review my main concern was how much this study really adds to our understanding of Efa6 function in the nervous system. The protein was already known to limit MT polymerization at the plasma membrane and the microtubule elimination domain (MTED) was previously shown to be critical for this function, both in non-neuronal cells and neurons. The authors have now added new data (Figure 4, Figure 6G and Figure 8N). Importantly, Figure 4 shows that the MTED peptide is capable of limiting MT polymerization in vitro when only tubulin and MTED peptide is present. Furthermore, they show that the MTED peptide is able to immunoprecipitate purified tubulin. These experiments suggest that the MTED can function directly with tubulin. These data do provide some mechanism that was lacking in the previous version of the manuscript. However, it is somewhat crude to compare MTED inhibition of polymerization to no addition of MTED (in the growth assay in Figure 4A) or by comparing MTED-coupled beads to empty beads (in the IP in Figure 4B). These are very weak negative controls, given that tubulin readily interacts non-specifically with other proteins in many assays and there are not any positive controls in the assays they perform in Figure 4. In Figure 4—figure supplement 1A and B they use MCAK in their in vitro assays as a positive control. Since the Efa6-Δ-Cterminal peptide does not seem to work in Figure 4—figure supplement 1A/B (the authors suggest this may be due to large disordered regions in that part of the protein), some other (scrambled?) peptide could be used as a negative control and/or MCAK (or peptide thereof) as a positive control. Such experiments would provide a more compelling mechanism with relatively little additional work. Additionally, the authors have also modified the previous manuscript to make it more coherent and compact and addressed my previous minor concerns.

As mentioned above, all the requested controls are now provided in Figure 4: A scrambled form of the peptide has no impact on microtubule growth and does not pull down tubulin when attached to beads. MCAK as a positive control also potently inhibits microtubule growth.

We also attached MCAK to beads by the same method used for the peptides, which is covalent attachment via N-terminal amine or lysine side chains. Since the peptides contain no lysine they are attached via the N-terminal amine. MCAK, attached to beads in this way, brings down very little tubulin and does not show the nucleotide-dependent binding previously observed by several labs (see Figure Author response image 1 and Wagenbach et al., 2008). This behaviour is likely due to the method of attachment. MCAK may be denatured by the conditions required for covalent attachment to CNBr-activated beads and/or may be bound in an orientation that masks the tubulin binding site which contains several lysine residues. When MCAK is attached to beads in a more controlled way (e.g. to Protein A coated beads via an anti-his tag antibody, Figure 1) it pulls down tubulin when in the ATP-bound state as mimicked by the nonhydrolysable ATP analogue AMPPNP. There is no peptide of MCAK currently known to bind tubulin, the interaction via the major tubulin binding site is nucleotide-dependent and requires the full motor domain.

**Author response image 1. respfig1:** Tubulin pull down by MCAK attached to sepharose beads using different methods of attachment. (**A**) MCAK covalently attached to beads via any lysine or N-terminal amine (Figure 4 of this publication). (**B**) MCAK attached to Protein A coated beads via anti-his antibody binding the C-terminal his-tag (Claire Friel, unpublished data).

Reviewer #3:This work is a comprehensive study on the function and molecular mechanism of efa6 in regulating MT organization during axonal growth and axonal maintenance. It uses a combination of systems to conclude that Drosophila efa6 is a cortical MT collapse factor blocking growth of MTs that have left the axon bundle (but that its function is not conserved in vertebrates). The authors have answered most of the points I raised in a previous review, and the manuscript has improved greatly, especially regarding the possible mechanism of efa6 action.The only comment is that the order of Figures is slightly mixed, as in it would be better to start the results with Figure 1A, instead of Figure 1F, and the first supplement. I encountered was Figure 3—figure supplement 1B, instead of Figure S1(now Figure 3—figure supplement 5.

As mentioned above, we have now corrected all figure issues to the best of our knowledge.

In addition, there are various typos and unclear sentences, e.g., subsection “The MTED is a good predictor of MT-inhibiting function directly affecting MT polymerisation”.

We very carefully read to the text and improved formulations in various positions. For example, the mentioned paragraph now reads: "Taken together, our structure-function data clearly establish *Drosophila* Efa6 as a cortical collapse factor: its N-terminal MTED blocks MT polymerisation, and this function is restricted to the cortex through the Efa6 C-terminus which associates with the cell membrane."